# When Do MLPs Excel in Node Classification? An Information-Theoretic Perspective

## Abstract

Recent research has shed light on the competitiveness of MLP-structured methods in node-level tasks. Nevertheless, there remains a gap in our understanding regarding why MLPs perform well and how their performance varies across different datasets. This paper addresses this lacuna by emphasizing mutual information's pivotal role in MLPs vs. GNNs performance variations. We first introduce a tractable metric to quantify the mutual information between node features and graph structure, based on which we observe different characteristics of various datasets, aligning with empirical results. Subsequently, we present InfoMLP, which optimizes node embeddings' mutual information with the graph's structure, i.e., the adjacency matrix. Our info-max objective comprises two sub-objectives: the first focuses on non-parametric reprocessing to identify the optimal graph-augmented node feature matrix that encapsulates the most graph-related information. The second sub-objective aims to enhance mutual information between node embeddings derived from the original node features and those from the graph-augmented features. This integration of message-passing during preprocessing maintains the efficiency of InfoMLP, ensuring it remains as efficient as a standard MLP during both training and testing. We validate the effectiveness of our approach through experiments on real-world datasets of varying scales, supplemented by comprehensive ablation studies. Our results affirm our analysis and underscore the success of our innovative approach.

## 1 Introduction

Learning representations and subsequently predicting node labels for nodes in an attributed graph is a fundamental task in graph machine learning that has attracted significant attention over the past decade. To learn from both node features and graph structure, Graph Neural Networks (GNNs) (Kipf & Welling, 2017; Velickovic et al., 2018; Hamilton et al., 2017; Xu et al., 2019; Chen et al., 2020) adopt an iterative process, aggregating messages from neighboring layers. By stacking multiple layers, GNNs can effectively learn node representations that capture information from both the node features and the local/global graph structure.

While Graph Neural Networks (GNNs) demonstrate promising performance, their message-passing scheme is often criticized for its inefficiency and time-consuming nature during both training and inference (Rossi et al., 2020; Zhang et al., 2021b; Zeng et al., 2020). Furthermore, its dependence on graph structure hinders its application in cold-start scenarios (Zheng et al., 2021). In contrast, Multilayer Perceptrons (MLPs) utilizing only node features as input offer efficiency during both training and inference. However, their performance on node classification tasks falls short due to the lack of graph structure information. Several attempts have been made to incorporate graph structure information into MLP models from different perspectives: 1) Regularization-based methods: These train an MLP encoder using a combination of supervised cross-entropy loss and a regularization loss concerning the graph structure (Ando & Zhang, 2006; Yang et al., 2021; Hu et al., 2021; Dong et al., 2022). The effectiveness of such methods often hinges on how the regularization loss is designed, as a well-designed regularization loss can more effectively integrate structure information into node embeddings. 2) Distillation-based methods: These leverage the KL-divergence loss to distill the predictions of a GNN teacher into an MLP student (Zhang et al., 2021b; Zheng et al., 2021), with the goal that the MLP can generate predictions similar to its GNN teacher. While these methods are comparable to a vanilla MLP in testing, they still require message passing during training to learn from

Table 1: Categorization of GCN and typical MLP-architected models. $X$ and $A$ denote the node features and graph adjacency matrix, respectively. We denote a model as $\mathcal{O}(\texttt{GCN})$ if it requires any form of message passing and as $\mathcal{O}(\texttt{MLP})$ if it does not. The $\text{MLP}_{\texttt{both}}$ symbol indicates that the model shares the same complexity as an MLP during both training and testing. The $\text{MLP}_{\texttt{test}}$ symbol represents a model that aligns with an MLP in terms of testing complexity but not during training.

| Model | Input | | Algorithmic Complexity | | | Category |
|---|---|---|---|---|---|---|
| | Training | Testing | Pre-processing | Training | Testing | |
| GCN (Kipf & Welling, 2017) | $(X, A)$ | $(X, A)$ | - | $\mathcal{O}(\texttt{GCN})$ | $\mathcal{O}(\texttt{GCN})$ | GNN |
| MLP | $X$ | $X$ | - | $\mathcal{O}(\texttt{MLP})$ | $\mathcal{O}(\texttt{MLP})$ | $\text{MLP}_{\texttt{both}}$ |
| Lap-Reg (Ando & Zhang, 2006) | $(X, A)$ | $X$ | - | $\mathcal{O}(\texttt{GCN})$ | $\mathcal{O}(\texttt{MLP})$ | $\text{MLP}_{\texttt{test}}$ |
| GraphMLP (Hu et al., 2021) | $(X, A)$ | $X$ | - | $\mathcal{O}(\texttt{GCN})$ | $\mathcal{O}(\texttt{MLP})$ | $\text{MLP}_{\texttt{test}}$ |
| N2N (Dong et al., 2022) | $(X, A)$ | $X$ | - | $\mathcal{O}(\texttt{GCN})$ | $\mathcal{O}(\texttt{MLP})$ | $\text{MLP}_{\texttt{test}}$ |
| GLNN (Zhang et al., 2021b) | $(X, A)$ | $X$ | - | $\mathcal{O}(\texttt{GCN})$ | $\mathcal{O}(\texttt{MLP})$ | $\text{MLP}_{\texttt{test}}$ |
| **InfoMLP** (ours) | $(X, A)$ | $X$ | $\mathcal{O}(KEd)$ | $\mathcal{O}(\texttt{MLP})$ | $\mathcal{O}(\texttt{MLP})$ | $\text{MLP}_{\texttt{both}}$ |

the graph structure. Table 1 categorizes GNNs and different MLP-architectured methods according to the input information and time complexity at training/testing phases. Despite the generally good performance of these MLP models, we observe clear performance discrepancies on different datasets, and the reasons behind these phenomena aren't fully understood.

In this work, we first aim to understand the reasons behind the successes of previous MLP-structured models for learning node representations. To achieve this, we propose a tractable measure to quantify the overlapping information between node features (the sole input of MLPs) and the graph structure (which plays a role in GNNs). Our proposed measure reveals a substantial overlap of information between node features and the graph structure. This observation implies that the node features, which partially capture the graph structure information, can be effectively learned by MLPs if appropriately trained. Motivated by this insight, we propose maximizing the mutual information between node embeddings and the graph structure as an auxiliary loss function for learning an MLP encoder. To realize this objective, we introduce **InfoMLP**. InfoMLP decomposes the mutual information maximization objective into two steps: 1) the generation of a graph-augmented node feature matrix that encapsulates extensive graph structure information, and 2) the maximization of mutual information between node representations learned from the original node features and the generated graph-augmented node features. By defining a non-parametric graph-augmented node-feature matrix, the first step becomes a dataset-specific preprocessing step. Consequently, InfoMLP maintains the efficiency of an MLP during both training and testing (as demonstrated in Table 1). We conduct experiments on graphs of various scales for node classification tasks in transductive, inductive, and cold-start settings. Our empirical results confirm the effectiveness, efficiency, and rationality of our proposed design. We summarize our contributions as follows:

1) We introduce a tractable measure to investigate the correlation between raw node features and the graph structure. Utilizing this measure, we can estimate the extent to which node features cover graph structure information. Our analysis aligns with prior empirical observations regarding the performance gaps between MLPs and GNNs.

2) We propose InfoMLP, a novel regularization-based MLP model designed for learning node representations on graphs. InfoMLP aims to maximize the mutual information between node embeddings and the graph structure. By decomposing the primary objective into a preprocessing step and a learning step, InfoMLP achieves the same efficiency as a standard MLP model during both the training and testing phases.

3) We perform experiments on graphs of varying sizes across three distinct settings. The empirical results confirm the effectiveness and efficiency of InfoMLP. Additionally, extensive ablation studies support the validity of our analysis and the rationale behind our designs.

## 2 RELATED WORKS: LEARNING MLPS ON GRAPHS.

Due to their universal approximation ability (Hornik et al., 1989), Multilayer Perceptrons (MLPs) play a critical role in a variety of machine learning tasks. However, MLPs are designed to handle independent and identically distributed (i.i.d.) data points and thus struggle with non-i.i.d. data,

such as nodes in a graph. Consequently, the performance of MLPs often falls short when compared with Graph Neural Networks (GNNs). To incorporate the knowledge of graph structure into the embeddings produced by MLPs, regularization-based methods employ an additional regularization loss alongside the supervised loss. This approach encourages the node embeddings to conform to the graph structure. Classic examples are Laplacian Regularization (Ando & Zhang, 2006; Zhou et al., 2003) and P-Reg (Yang et al., 2021), which promotes Laplacian-smoothed and Propagation-smoothed predictions of node labels. Although they can capture some aspects of structure information, their performance improvements are relatively modest, and they still can't compete with GNNs.

In recent years, advanced MLP models have emerged, achieving comparable performance to GNNs on certain datasets. For example, GraphMLP (Hu et al., 2021) employs a contrastive loss that considers connected nodes as positive pairs, while N2N (Dong et al., 2022) maximizes the mutual information between node embeddings and sampled neighborhood embeddings. GLNN (Zhang et al., 2021b) utilizes knowledge distillation to encourage MLP students to produce GNN-like predictions. Although these advanced methods have greatly narrowed the gaps between the performance of MLPs and GNNs, it is observed that they exhibit varying performance on distinct datasets, and the reasons behind these phenomena are still under-explored.

It is worth noting this paper focuses on MLPs that take pure raw node attributes as input for obtaining node representations without the graph structure information (i.e., at the testing phase, the mode is supposed to take merely $X$ rather than $(X, A)$ as input, see Table 1). Therefore, we would like to clarify that our research topic differs from the following works: 1) Graph-augmented MLPs (e.g., SGC (Wu et al., 2019) and SIGN (Rossi et al., 2020)) are significantly different since they apply message passing to either the MLPs' input or output, explicitly utilizing the graph structure information to generate node embeddings. 2) Training-time MLP methods (e.g., PMLP (Yang et al., 2023) and MLP-init (Han et al., 2023)) do not utilize the graph structure information in training, but still apply message passing in testing. 3) Another GNN-to-MLP knowledge distillation method NOSMOG (Tian et al., 2022) takes structural embeddings (from DeepWalk, (Perozzi et al., 2014)) as additional inputs of the MLP model, therefore still utilizing the graph structure information.

## 3 METHODOLOGY

**Preliminaries.** This paper studies a general node representation learning task for node classification in both transductive and inductive settings. For a graph $\mathcal{G}$ comprising $N$ nodes and $E$ edges, we're given two associated sources of information: raw node feature matrix $X \in \mathbb{R}^{N \times d}$ and graph structure matrix $A \in \mathbb{R}^{N \times N}$, where $d$ denotes the dimension of raw node features. The difference between the transductive and inductive settings lies in whether the testing nodes and their associated edges are observable during the training phase. The objective is to learn informative node representations, denoted by $Z \in \mathbb{R}^{N \times D}$, where $D$ is the embedding dimension. Graph neural networks (GNNs) accomplish this by learning an encoder that takes both $X$ and $A$ as inputs, yielding $Z_{\text{gnn}} = \text{GNN}(X, A)$. In contrast, MLPs learn an encoder using only the raw features $X$, resulting in $Z_{\text{mlp}} = \text{MLP}(X)$. Regardless of the encoder used, the training process is steered by a task-specific objective function, such as the cross-entropy loss function computed between the predicted and ground-truth labels $Y$. This can be construed as maximizing the mutual information between predictions and the ground truth labels: $\max I(Z, Y)$.

### 3.1 GAPS BETWEEN MLPS AND GNNS FOR NODE REPRESENTATION LEARNING

Given the differing input sources, there is a considerable performance disparity between MLPs and GNNs when applied to the same dataset. As depicted in Table 2, the vanilla MLP tends to underperform GCN significantly. This performance gap is typically ascribed to the input differences: GNNs utilize both node features $X$ and the graph structure $A$, while MLPs depend solely on node features $X$. Nonetheless, recent studies have shown that with suitable regularization or training strategies, the performance of MLPs can be boosted to match, or even surpass,

Table 2: Performance comparison of GCN and MLP-architectured models on several datasets. The results are obtained in the transductive setting (the same as in Table 3).

| Dataset | Cora | Citeseer | Pubmed | Computer | CS |
|---|---|---|---|---|---|
| GCN | 81.5 | 70.3 | 79.0 | 92.25 | 94.10 |
| MLP | 59.7 | 57.1 | 68.4 | 85.42 | 95.97 |
| $\triangle_{\text{GCN}}$ | −26.7% | −18.8% | −13.4% | −7.40% | +1.98% |
| GraphMLP | 79.5 | 73.1 | 79.7 | 88.73 | 96.62 |
| $\triangle_{\text{GCN}}$ | −2.5% | +4.0% | +0.9% | −3.82% | +2.68% |
| N2N | 82.8 | 73.3 | 80.7 | 89.07 | 96.41 |
| $\triangle_{\text{GCN}}$ | +1.6% | +4.3% | +2.2% | −3.45% | +2.45% |

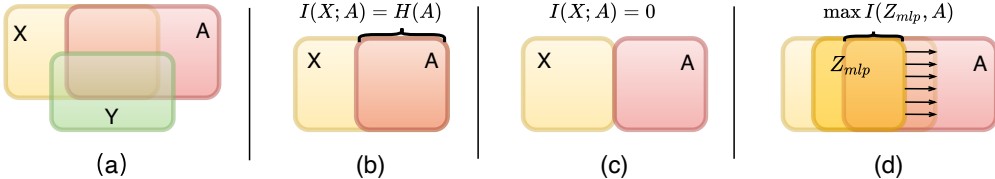

Figure 1: An illustration of the relationships between the information of node features $X$, the graph structure $A$, embeddings of an MLP encoder $Z_{\texttt{mlp}}$, and the downstream task $Y$. (a) Both $X$ and $A$ contains informative information for the downstream task. (b) the case that $X$ contains all the information about $A$. (c) the case that $X$ contains no information about $A$. (d) maximizing the mutual information between the embeddings $Z_{\texttt{mlp}}$ and the graph structure $A$.

GNNs on certain datasets. As seen in Table 2, GraphMLP (Hu et al., 2021) and N2N (Dong et al., 2022) exhibit comparable, or superior, performance to GCN on four out of five datasets. We hypothesize that this is due to the high degree of overlap between the information conveyed by node features $X$ and the graph structure $A$, which is very common in real-world graphs such as social networks. We illustrate this point using the following assumption of the generation process of data:

**Assumption 1.** *(Generation process of graph) For a graph $\mathcal{G}$ of node attributes $X$ and adjacency matrix $A$, the raw node features $X$ for the nodes is generated first, then the graph structure $A$ is generated based on the node features and additional confounder factors $F$. Finally, the node labels $Y$ are generated from both the raw node features $X$ and the graph structure $A$. Formally:*

$$p(A, X) = p(X)p_F(A|X), \qquad p(A, X, Y) = p(X, A)p(Y|A, X) \qquad (1)$$

Since node labels $Y$ can be inferred from $X$ and $A$, and $A$ itself is generated from $X$ and another confounder factor, if X already contains enough information about A, then an MLP model can achieve performance similar to GNN with X as input alone. Specifically, when the label $Y$ is solely generated using $X$, (e.g., $A$ is simply noise), an MLP model naturally outperforms GNN.

We can comprehend the intuition behind the overlap through the Venn diagrams depicted in Fig. 1. Fig. 1(a) illustrates that both $X$ (node features) and $A$ (graph structure) harbor information pertinent to the downstream tasks $Y$. There is also an overlapping section between $X$ and $A$, which may contain substantial information beneficial for the downstream task $Y$. Fig. 1(b) presents a scenario where all information from the graph structure is contained within the node features, i.e., $I(X; A) = H(A)$ or $H(A|X) = 0$. In such a scenario, the graph structure $A$ is entirely predictable using $X$. Thus, MLPs are expected to have the same generalization capacity as GNNs in this context. On the other hand, Fig. 1(c) displays a situation where the graph structure is unrelated to the node features, i.e., $I(X; A) = 0$. In this case, regulating $Z_{\texttt{mlp}}$ with the graph structure information does not assist in improving the node embeddings to recover the graph structure. Consequently, the performance of MLPs cannot be enhanced in such a scenario.

Under the assumption of graph homophily (McPherson et al., 2001; Ciotti et al., 2016), the graph structure information is presumed to contain valuable data for predicting node labels (the downstream task $Y$). As such, if the mutual information between node representations and the graph structure $I(Z_{\texttt{mlp}}; A)$ could be maximized, $Z_{\texttt{mlp}}$ is expected to encompass the most pertinent information for downstream tasks, as illustrated in Fig. 2 (d). Although conventional methods like Lap-Reg (Ando & Zhang, 2006) and P-Reg (Yang et al., 2021) aim to learn smoothed node representations/predictions, they fall short in maximizing $I(X_{\texttt{mlp}}; A)$ due to the absence of negative terms. In contrast, newer methods such as GraphMLP (Hu et al., 2021) and N2N (Dong et al., 2022) take into account both actual edges and unconnected node pairs, thereby striving to reconstruct the complete adjacency matrix. In doing so, the mutual information between embeddings $X_{\texttt{mlp}}$ and $A$ is implicitly maximized, which makes these methods more effective.

The analysis above identifies two crucial factors determining the gap between MLPs and GNNs: 1) **The mutual information between the node attributes and the graph structure $I(X; A)$.** 2) **The mutual information between the embeddings and the graph structure $I(Z_{\texttt{mlp}}; A)$.** The former is dataset-deterministic and establishes the maximum capacity of $Z_{\texttt{mlp}}$, while the latter can be improved

through the design of the model. In Sec. 3.3, we propose a novel regularization-based MLP model, which is able to maximize $I(Z_{\texttt{mlp}}, A)$ more straightforwardly.

## 3.2 Quantifying the Overlapped Information between Features and Structure

Mutual information can be deconstructed into the difference between entropy and conditional entropy: $I(X; A) = H(A) - H(A|X)$. Instead of focusing on the information inherent to the graph structure, we can examine the conditional term $H(A|X)$. This term signifies the residual information about the graph structure given the observation of node features. A small $H(A|X)$ value suggests that the graph structure $A$ can be effectively predicted from node features $X$. However, quantifying $H(A|X)$ presents a challenge due to the high dimensionality of vector $X$ and the large sparse nature of matrix $A$. Fortunately, we can estimate $H(A|X)$ by leveraging a computationally feasible upper bound:

**Theorem 1.** *Let $\hat{A}$ be a function of $X$, i.e., $\hat{A} = f(X) \in \mathbb{R}^{N \times N}$, then $H(A|X)$ is upper bounded by $H(A|\hat{A})$. Formally, $H(A|X) \le \inf_f H(A|\hat{A})$.*

We can strategically devise a suitable function $\hat{A} = f(X)$ to estimate $H(A|X)$ using its upper limit $H(A|\hat{A})$. In this paper, we define it as the squared distance between the $\ell_2$ normalized features: $\hat{A}_{ij} = \ell_2^2(\hat{\boldsymbol{x}}_i, \hat{\boldsymbol{x}}_j) = \|\hat{\boldsymbol{x}}_i - \hat{\boldsymbol{x}}_j\|_2^2$, motivated by the widely adopted graph homophily assumption that nodes with similar features tend to formulate edges. Subsequently, $H(A|X)$ can be upper-bounded by $H(A|\hat{A})$:

$$H(A|X) \le H(A|\hat{A}) = -\sum_{i=1}^{N}\sum_{j=1}^{N} \log p(A_{ij}|\hat{A}_{ij}). \tag{2}$$

In Theorem 2, we demonstrate an upper bound of the gap between $H(A|X)$ and $H(A|\hat{A})$ where $\hat{A}$ is estimated using the $\ell_2$ distance:

**Theorem 2.** *Assume the $\ell_2$ distance of positive (real-existing) edges follows a Gaussian distribution $p_{\text{pos}} \sim \mathcal{N}(\mu_p, \sigma_p^2)$, and the $\ell_2$ distance of negative (non-existing) edges follows another Gaussian distribution $p_{\text{neg}} \sim \mathcal{N}(\mu_n, \sigma_n^2)$. Let $\mu_m = \frac{1}{2}(\mu_p + \mu_n)$, and $\sigma_m^2 = \frac{1}{2}(\sigma_p^2 + \sigma_n^2)$. With the assumption that the probability that an edge is positive/negative is the same, the gap between $H(A|X)$ and $H(A|\hat{A})$ is*

$$H(A|X) - H(A|\hat{A}) = I(A; X) - \mathcal{D}_{JS}(p_{\text{pos}}\|p_{\text{neg}}), \tag{3}$$

*where $\mathcal{D}_{JS}$ denotes the Jensen-Shannon divergence, and*

$$\mathcal{D}_{JS}(p_{\text{pos}}\|p_{\text{neg}}) = \frac{1}{2}\left(\frac{(\mu_p - \mu_m)^2 + (\mu_n - \mu_m)^2 + \sigma_p^2 + \sigma_n^2}{\sigma_m^2} - 2 + \log\left(\frac{\sigma_m^4}{\sigma_p^2\sigma_n^2}\right)\right) \tag{4}$$

Theorem 2 states that the gap between $H(A|X)$ and $H(A|\hat{A})$ is influenced by the difference between the distributions of positive edges and negative edges. Specifically, when the difference between these two distributions is larger, the gap becomes smaller, and using $H(A|\hat{A})$ to estimate $H(A|X)$ becomes more accurate.

Considering that $A_{ij} = 0/1$ represents the existence of an edge between nodes $i$ and $j$, and $\hat{A}_{ij}$ is a scalar, we can estimate $p(A_{ij}|\hat{A}_{ij})$ using kernel density estimation (We defer the detailed steps in Appendix C). This involves estimating the negative distribution $p(\hat{A}_{ij}|A_{ij} = 0)$ and the positive distribution $p(\hat{A}_{ij}|A_{ij} = 1)$. We apply this method to the five datasets in Table 2, and the results are presented in Fig. 2. Combining the results in Table 2, we observe three different scenarios:

**1) Small entropy**: For CS , the positive and negative distributions are easy to distinguish, resulting in the smallest $H(A|\hat{A})$. Most of the graph structure information is already captured within the node features, explaining why the vanilla MLP obtains competitive performance on CS .

**2) Medium entropy**: For Cora, Citeseer, and Pubmed, there is a moderate overlap between the two distributions, indicating that graph structure can be predicted to some extent using node features. In this case, although the vanilla MLP's performance is poor, advanced MLPs with proper designs (e.g., GraphMLP and N2N in Table 2 can obtain competitive performance close to GNNs.

Figure 2: Estimated probability density function of $p(\hat{A}|A)$ and conditional entropy $H(A|\hat{A})$. Blue line/curve stands for real edges ($A_{ij} = 1$) while orange line/curve stands for non-existing ones ($A_{ij} = 0$). The divergence between the two distributions intuitively hints at how much structural information is covered in the node features: the larger the divergence, the more easily real edges and non-existing edges can be distinguished using $\hat{A}$, then the smaller $H(A|\hat{A})$ and $H(A|X)$.

**3) Large entropy**: For `Computer`, the positive and negative distributions are heavily overlapped, indicating the difficulty of discerning edge presence based on node features alone. Consequently, MLP methods' performance on `Computer` is still inferior to GCN, even with proper regularization.

These observations align with the analysis in Sec. 3.1 that the performance gap between MLPs and GNNs is constrained by $I(X, A)$ – when node attributes contain little graph structure information, the performance of the MLP model can hardly be improved to the level of GNNs. Considering that the node attribute matrix $X \in \mathbb{R}^{N \times d}$, while the graph adjacency matrix $A \in \mathbb{R}^{N \times N}$, the gap between the dimension $d$ and graph size $N$ directly affect how much information $X$ can convey about $A$. Specifically, when $N$ overwhelms $d$, attempting to infer the graph structure from node features becomes difficult. This is consistent with what Figure 2 shows: `Computer` and `CS` datasets have similar sizes, but `CS` has node features of nearly $6,800$ dimensions, while `Computer`'s is slightly over 700 dimensions. Therefore `CS` can better capture the graph structure using its node features.

### 3.3 LEARNING EFFECTIVE MLPs VIA MUTUAL INFORMATION MAXIMIZATION

The above analysis and observation outline that when the graph structure harbors significant information for predicting node labels, the ideal node representations $Z_{\texttt{mlp}}$ should encapsulate as much information about the graph structure as possible. Prior methods (Hu et al., 2021; Dong et al., 2022; Zhang et al., 2021b) have implicitly approached this target and achieved notable performance. Their success inspires us to devise a method that can directly address this target in a more explicit fashion. In this section, we introduce InfoMLP, a regularization-based MLP model explicitly designed for the task of mutual information maximization.

However, maximizing $I(Z_{\texttt{mlp}}; A)$ presents a challenge due to the following reasons: 1) $Z_{\texttt{mlp}}$ and $A$ have distinct shapes; 2) $A$ can become a very large matrix when the graph size is large. To overcome these challenges, we need an implementation- and optimization-friendly approach. To this end, we leverage a graph-augmented node feature matrix $X_{\texttt{aug}} = g(X, A) \in \mathbb{R}^{N \times D}$, which assigns an auxiliary feature vector $\boldsymbol{x}_{\texttt{aug},i} \in \mathbb{R}^D$ to each node $i$. This allows us to maximize $I(Z_{\texttt{mlp}}; A)$ via the following two steps: 1) Identifying an optimal graph-augmented node feature matrix $X_{\texttt{aug}} = g(X, A)$ such that $H(A|X_{\texttt{aug}})$ is minimized; 2) Training an MLP encoder $Z_{\texttt{mlp}} = \texttt{MLP}_\theta(X)$ such that $I(Z_{\texttt{mlp}}; X_{\texttt{aug}})$ is maximized. We now elaborate on how these two steps can be implemented in a simple and efficient manner.

**Minimization of** $H(A|X_{\texttt{aug}})$. The design of $X_{\texttt{aug}} = g(X, A)$ can be quite flexible. For instance, we could choose $g(X, A)$ to be a neural network with an extensive number of learnable parameters. However, optimizing the neural network can be challenging as estimating $H(A|X_{\texttt{aug}})$ (as detailed in Section 3.2) is a non-differentiable process. Therefore, we consider a **non-parametric** implementation of $g$ via the generalized graph diffusion (Gasteiger et al., 2019):

$$X_{\texttt{aug}}(K) = g(X, A) = \sum_{k=1}^{K} \gamma_k \tilde{A}^k X, \tag{5}$$

where $\tilde{A}$ represents the symmetrically normalized graph adjacency matrix, $\gamma_k$ is predefined weighting coefficients. One may assign $\gamma_k$ arbitrarily to make $X_{\text{aug}}$ exhibit different properties. We simply set $\gamma_k = 1/K$ with which satisfying performance could be obtained.

By using this formulation, the optimal $X_{\text{aug}}$ can be obtained by selecting the $K$ with the lowest conditional entropy $H(A|X_{\text{aug}}(K))$. This design offers the following advantages: 1) The process is non-parametric, allowing for preprocessing prior to the training of the remaining models. 2) Although there is a hyperparameter $K$, it is unrelated to the optimization process. As a result, the optimal value for $K$ can be determined without training the model, and it can be selected by evaluating the performance on the validation set.

**Maximization of** $I(Z_{\text{mlp}}; X_{\text{aug}})$ Although $Z_{\text{mlp}}$ and $X_{\text{aug}}$ can be considered as stacks of i.i.d. vectors, and there exists a correspondence (e.g., $z_{\text{mlp},i}$ and $x_{\text{aug},i}$ correspond to the same node $i$), directly computing $I(Z_{\text{mlp}}; X_{\text{aug}})$ is inconvenient due to their different dimensions. Fortunately, we can still maximize a tractable lower bound $I(Z_{\text{mlp}}; Z_{\text{aug}})$ instead:

$$I(Z_{\text{mlp}}; X_{\text{aug}}) \geq I(Z_{\text{mlp}}; Z_{\text{aug}}), \qquad (6)$$

where $Z_{\text{aug}} = \text{MLP}_\theta(X_{\text{aug}})$ using the same MLP encoder as $Z_{\text{mlp}}$. Finally, we can maximize $I(Z_{\text{mlp}}; Z_{\text{aug}})$ using well-studied mutual information estimators (Belghazi et al., 2018; van den Oord et al., 2018). For the complexity consideration, we adopt the following MI maximizer based on feature decorrelation (Zhang et al., 2021a) (a detailed introduction in Appendix B.3):

$$\mathcal{L}_{\text{MI}} = -\alpha \sum_{i=1}^{N} \|\tilde{z}_i^{\text{mlp}} - \tilde{z}_i^{\text{aug}}\|_2^2 + \beta \sum_{p \neq q} C_{pq}^2, \qquad (7)$$

where $z_i$ is the $i$-th row of $\tilde{Z}$, and $\tilde{Z}$ is the standardization of $Z$ along the instance dimension – $\tilde{Z} = Z - \mu(Z)/(\sigma(Z) \cdot \sqrt{N})$. $C \in \mathbb{R}^{D \times D}$ is the auto-correlation matrix of $Z_{\text{mlp}}$, and is computed as $C = \tilde{Z}^\top \tilde{Z}$. $\alpha$ and $\beta$ are two trade-off hyperparameters. Notably, the computation complexity of Eq. 7 is $\mathcal{O}(ND^2)$ (whereas the popular InfoNCE MI estimator (van den Oord et al., 2018) is $\mathcal{O}(N^2D)$), which is the same as a linear layer in MLP. This indicates that computing Eq. 7 will not add extra computational burden.

**Model Training**. Having discussed the underlying components, we can now provide an overview of how our model is trained. The training process consists of two steps: Step 1) Selecting the optimal graph-augmented node feature matrix $X_{\text{aug}}$ to minimize $H(A|X_{\text{aug}})$. This step does not involve learning and can be completed by conducting several trials and comparisons, as we have specified $X_{\text{aug}}$ with Eq. 5. It is important to note that Step 1 is non-learning and can be fully processed ahead of Step 2. Step 2) Maximizing the mutual information between the two embedding matrices $Z_{\text{mlp}}$ and $Z_{\text{aug}}$ encoded using the same MLP encoder $\text{MLP}_\theta$. Furthermore, the node embeddings $Z_{\text{mlp}}$ are utilized to predict the node labels. The objective function for Step 2 is as follows:

$$\min_{\theta,\phi} \mathcal{L} = \sum_{i \in \mathcal{V}^L} \text{CE}(\hat{y}_{\text{mlp},i}, y_i) + \mathcal{L}_{MI}, \qquad (8)$$

where CE refers to Cross-Entropy. $\mathcal{V}^L$ is the labeled node set, $y_i$ is the ground-truth label, and $\hat{y}_{\text{mlp},i} = \text{FC}_\phi(z_{\text{mlp},i})$ is the predicted logits for node $i$. $\text{FC}_\phi$ is the linear prediction head with learnable parameters $\phi$. We provide a pseudo-code of InfoMLP in the Appendix A.

**Model Testing**. During the testing phase, InfoMLP operates in the same manner as a vanilla MLP. Given a testing node of feature $x$, InfoMLP predicts its label using $\hat{y} = \text{FC}_\phi(\text{MLP}_\theta(x))$.

**Complexity**. The complexity of InfoMLP can be analyzed for its preprocessing, training, and testing steps separately. The preprocessing step of InfoMLP involves the computation of the graph-augmented feature matrix, which is propagated $K$ times. This step has a complexity of $\mathcal{O}(KEd)$. During the training of InfoMLP, the encoding of an MLP, mutual information maximization, and linear prediction are involved. If we denote the complexity of an MLP model as $\mathcal{O}(\text{MLP})$, the overall complexity of InfoMLP's training can be expressed as $\mathcal{O}(\text{MLP}) + \mathcal{O}(\text{MI}) = \mathcal{O}(\text{MLP})$. For the testing phase, InfoMLP behaves like a vanilla MLP, resulting in a complexity of $\mathcal{O}(\text{MLP})$. Hence, except for the preprocessing step (which is performed once), **InfoMLP has the same complexity as a vanilla MLP in both training and testing**.

## 4 EXPERIMENTS

We conduct extensive experiments to validate the effectiveness and efficiency of the proposed InfoMLP model. In Section 4.1, we present the results of node classification on seven medium-sized graphs. Additionally, in Section E.1, we conduct further experiments to demonstrate the rationale behind the design choices. To provide a comprehensive analysis, we also include the ablation studies, the additional experimental results on large-scale graphs, heterophilic graphs, et.c., in Appendix E. These results further support the efficacy and efficiency of the proposed InfoMLP model.

### 4.1 NODE CLASSIFICATION ON MEDIUM-SIZED GRAPHS

**Experimental settings**. We consider three settings for evaluating InfoMLP's performance on node classification tasks: 1) the traditional **transductive node classification** setting, where all nodes and edges are available in training and testing; 2) the less-studied **inductive node classification** setting, where validation/testing nodes are not presented in training, and their connections to existing nodes will be utilized for making predictions; 3) an under-explored and much more challenged **inductive cold-start setting**, which is also inductive, but the connections of validation and testing nodes are not available during the inference stage.

**Datasets**. Following previous works (Zhang et al., 2021b; Hu et al., 2021), we consider 7 medium-sized graph datasets: Cora, Citeseer, Pubmed, Computer, Photo, CS, and Physics. The splits of these datasets are as follows: For Cora, Citeseer, and Pubmed, we directly use the public split, where 20 nodes are for training, 1500 nodes are for validation, and 1000 nodes are for testing. For Computer, Photo, CS, and Physics, no splits are provided, so we randomly split the nodes into training/validation/testing with 8:1:1 ratio. For all the experimental settings, we report the mean accuracy with standard deviation over 20 random trials.

**Baselines**. We compare InfoMLP with both GNNs and other MLP-related models that merely take node attributes as input. For GNNs, we consider SGC (Wu et al., 2019), GCN (Kipf & Welling, 2017), GAT (Velickovic et al., 2018), APPNP (Klicpera et al., 2019) and JK-Net (Xu et al., 2018). For previous MLPs, we compare with $MLP_{both}$: the vanilla MLP, and $MLP_{test}$: GraphMLP (Hu et al., 2021), GLNN (Zhang et al., 2021b), N2N (Dong et al., 2022), and NOSMOG (Tian et al., 2022). Note that the vanilla NOSMOG uses structural node embeddings as an additional input of the MLP and, as a result, is out of the scope of this paper. In our reproduction, we removed this component for a fair comparison. Furthermore, some knowledge distillation methods (Zhang et al., 2021b; Tian et al., 2022) directly distill testing nodes, leading to information leakage. We correct this error in our reproduction (see the discussion in Appendix D.4). The proposed InfoMLP has the same complexity as the vanilla MLP in both training and testing, so it also belongs to $MLP_{both}$.

**Results in the Transductive Setting**. Table 3 presents the results in the transductive setting. It is observed that our InfoMLP outperforms previous MLP-based models on all seven datasets. Furthermore, InfoMLP achieves better performance than competitive GNN models on six out of the seven datasets. These results highlight the effectiveness of utilizing mutual-information maximization as a regularization technique for improving MLP performance in learning node representations. Notably, Computer is the only dataset where InfoMLP falls short of beating GNN models. This can be attributed to the limited information gap between the raw node features and the graph structure, as illustrated in Fig. 2.

**Results in the Inductive and Cold-start Setting**. Table 4 presents the results in the inductive and cold-start settings. Several observations can be made from the results: 1) GNNs demonstrate advantages over MLPs in the inductive setting, where explicit utilization of the graph structure leads to improved performance. 2) In the challenging cold-start setting, GNNs experience a significant performance drop, and MLPs may even outperform them. 3) Our proposed InfoMLP consistently achieves superior performance in the cold-start setting, outperforming both GNNs and MLPs across all seven datasets. In the less favorable inductive setting, InfoMLP still achieves competitive or better performance, except for the Cora dataset, where the graph structure information appears to have more relevance to the node labels. These results validate the effectiveness of InfoMLP in learning informative node representations under various scenarios, including challenging cold-start situations.

Table 3: Mean and STD of testing accuracy in the transductive setting. **Bold Face** represents the best result on this dataset; **Bold Red** and **Bold Blue** represents the relative **improvement**/**decline** of InfoMLP compared with the best baseline, respectively.

| | Dataset | Cora | Citeseer | Pubmed | Computer | Photo | CS | Physics |
|---|---|---|---|---|---|---|---|---|
| GNNs | SGC | 81.0±0.5 | 71.9±0.5 | 78.9±0.4 | 89.92±0.37 | 94.35±0.19 | 94.00±0.30 | 96.19±0.13 |
| | GCN | 81.9±0.5 | 71.6±0.4 | 79.3±0.3 | **92.25±0.61** | 95.16±0.92 | 94.10±0.34 | 96.64±0.36 |
| | GAT | 83.0±0.7 | 72.5±0.7 | 79.0±0.3 | 91.72±0.85 | 95.05±0.98 | 94.19±0.31 | 96.71±0.14 |
| | JKNET | 81.3±0.5 | 69.7±0.2 | 78.9±0.6 | 91.25±0.76 | 94.82±0.22 | 93.57±0.49 | 96.31±0.29 |
| | APPNP | 82.6±0.2 | 71.7±0.5 | 80.3±0.1 | 91.81±0.78 | 95.84±0.34 | 94.41±0.29 | 96.84±0.26 |
| MLP$_{both}$ | Vanilla MLP | 59.7±1.0 | 57.1±0.5 | 68.4±0.5 | 85.42±0.51 | 92.91±0.48 | 95.97±0.22 | 96.90±0.27 |
| MLP$_{test}$ | GraphMLP | 79.5±0.6 | 73.1±0.4 | 79.7±0.4 | 88.73±0.58 | 95.68±0.28 | 96.62±0.43 | 97.04±0.16 |
| | N2N | 82.8±0.4 | 73.3±0.5 | 80.7±0.4 | 89.07±0.41 | 95.99±0.47 | 96.41±0.51 | 97.29±0.24 |
| | GLNN | 82.4±0.5 | 72.8±0.4 | 80.5±0.6 | 87.82±0.40 | 95.19±0.24 | 96.43±0.33 | 97.11±0.29 |
| | NOSMOG | 82.7±0.5 | 72.6±0.4 | 81.1±0.4 | 88.18±0.36 | 95.77±0.31 | 96.37±0.19 | 97.34±0.38 |
| MLP$_{both}$ | InfoMLP | **83.8±0.3** | **73.7±0.3** | **83.2±0.9** | 89.53±0.47 | **96.34±0.38** | **96.66±0.23** | **97.86±0.15** |
| | $\Delta_{BestGNN}$ | +0.96% | +1.65% | +3.61% | −2.94% | +0.52% | +2.17% | +1.19% |
| | $\Delta_{BestMLP}$ | +1.20% | +0.54% | +2.59% | +0.52% | +1.41% | +0.04% | +0.53% |

Table 4: Mean and STD of testing accuracy in the inductive and cold-start setting. Inductive/cold start makes no difference for MLP methods. Underline represents the best result in inductive setting, whereas **Bold Face** represents the best result in cold-start setting.

| | Dataset | Cora | Citeseer | Pubmed | Computer | Photo | CS | Physics |
|---|---|---|---|---|---|---|---|---|
| GNNs inductive | SGC | 74.2±3.6 | 67.4±0.3 | 71.5±0.3 | 89.89±0.26 | 93.25±0.34 | 93.65±0.26 | 96.38±0.21 |
| | GCN | 79.2±0.5 | 71.1±0.3 | 77.7±0.3 | 92.83±0.32 | 94.41±0.33 | 93.06±0.39 | 96.81±0.14 |
| | GAT | 81.1±0.8 | 71.4±0.4 | 77.2±0.9 | 91.88±0.29 | 94.30±0.37 | 93.82±0.23 | 96.68±0.18 |
| | JKNET | 81.0±0.6 | 67.3±1.1 | 77.9±0.6 | 90.97±0.88 | 93.25±0.76 | 92.41±0.09 | 96.42±0.08 |
| | APPNP | 81.1±0.9 | 70.8±0.4 | 79.6±0.5 | 92.85 ± 0.32 | 95.48 ± 0.13 | 94.46±0.17 | 97.03±0.13 |
| GNNs cold-start | SGC | 57.9±0.5 | 55.0±0.8 | 68.4±0.9 | 80.88±0.31 | 88.89±0.27 | 92.26±0.35 | 95.94±0.39 |
| | GCN | 65.9±0.3 | 64.2±0.4 | 74.8±0.5 | 82.58±0.13 | 88.36±0.25 | 93.78±0.43 | 96.55±0.18 |
| | GAT | 67.3±0.8 | 65.1±0.6 | 74.8±0.6 | 82.28±0.31 | 88.34±0.05 | 94.43±0.02 | 96.29±0.07 |
| | JKNET | 63.4±0.8 | 58.4±0.9 | 72.2±0.4 | 78.85±2.11 | 84.90±1.87 | 92.69±0.12 | 96.11±0.09 |
| | APPNP | 65.6±0.2 | 64.5±0.2 | 75.4±0.1 | 81.25±0.11 | 87.32±0.08 | 93.24±0.14 | 95.65±0.15 |
| MLP$_{both}$ | Vanilla MLP | 59.7±1.0 | 57.1±0.5 | 68.4±0.5 | 85.42±0.51 | 92.91±0.48 | 95.97±0.22 | 96.90±0.27 |
| MLP$_{test}$ | GraphMLP | 62.2±0.5 | 63.2±0.6 | 79.4±0.4 | 87.79±0.18 | 92.67±0.14 | 96.17±0.10 | 96.89±0.07 |
| | N2N | 63.4±0.4 | 62.9±0.5 | 78.2±0.4 | 87.56±0.13 | 92.51±0.19 | 96.28±0.09 | 96.99±0.18 |
| | GLNN | 62.8±0.5 | 63.7±0.5 | 79.1±0.5 | 86.94±0.23 | 91.57±0.18 | 95.90±0.34 | 96.19±0.12 |
| | NOSMOG | 64.2±0.6 | 65.2±0.3 | 79.2±0.4 | 87.22±0.19 | 92.42±0.23 | 96.11±0.26 | 96.74±0.17 |
| MLP$_{both}$ | InfoMLP | **68.9±0.4** | **70.1±0.7** | **82.2±0.4** | **88.37±0.09** | **92.94±0.13** | **96.80±0.11** | **97.22±0.18** |
| | $\Delta_{BestGNN-ind}$ | −15.04% | −1.82% | +4.52% | −4.82% | −2.66% | +2.47% | +0.19% |
| | $\Delta_{BestGNN-cold}$ | +2.38% | +7.68% | +10.34% | +7.01% | +4.56% | +2.51% | +0.69% |
| | $\Delta_{BestMLP}$ | +7.32% | +7.52% | +4.79% | +0.66% | +0.03% | +0.54% | +0.23% |

## 5 CONCLUSIONS

This paper has proposed InfoMLP, a novel MLP-structured model for node representation learning and node classification tasks on graphs. InfoMLP follows a mutual information maximization principle and is targeted to maximize the information between the embeddings of an MLP encoder and the graph structure. By employing a graph-augmented node feature matrix that captures as much of the graph structure information, InfoMLP achieves the same training/testing complexity as the vanilla MLP. Extensive experiments have verified the effectiveness of the proposed method.

Despite the superior performance and efficiency of InfoMLP, its performance highly depends on the quality of node features. As analyzed in Sec. 3.1, when the graph structure is much more important than the features, and the node features cover almost no information of the graph structure, MLPs performance can hardly be improved through structural regularization. Therefore, a promising direction is to study how to extract node features of high quality from the raw data.

## 6 REPRODUCIBILITY STATEMENT

We describe our algorithm in Appendix A. The proofs of our theoretical results are in Appendix B. The detailed implementations are provided in Appendix D, and the codes are provided in the supplementary.

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

# A ALGORITHM OF INFOMLP

We provide the pseudo-code of InfoMLP in Algorithm 1.

---
**Algorithm 1:** Algorithm for InfoMLP

---
**Input:** A graph $\mathcal{G} = (X, A) = (\mathcal{V}, \mathcal{E})$ with $N$ nodes and $E$ edges, where $X$ is node feature matrix, $A$ is the adjacency matrix. Ground-truth labels $Y^L$. An randomly initialized MLP encoder $\texttt{MLP}_\theta$, a randomly initialized linear classifier $\texttt{FC}_\phi$.
**Output:** Optimal parameters $\theta$ and $\phi$.

1 **Preprocessing**
2 **for** $K \in [1, K_{max}]$ **do**
3     $X_{\texttt{aug}}(K) = \sum_{k=1}^{K} \tilde{A}^k X / K$
4     Estimate $H(A|X_{\texttt{aug}}(K))$ according to Appendix C
5 $K^* = \arg\max H(A|X_{\texttt{aug}}(K))$
6 $X_{\texttt{aug}} = \sum_{k=1}^{K^*} \tilde{A}^k X / K^*$
7 **Training**
8 **while** *not converging* **do**
9     $Z_{\texttt{mlp}} = \texttt{MLP}_\theta(X), Z_{\texttt{aug}} = \texttt{MLP}_\theta(X_{\texttt{aug}})$
10     $\hat{Y} = \texttt{FC}_\phi(Z_{\texttt{mlp}})$
11     $\mathcal{L} = \texttt{CE}(\hat{Y}^L, Y^L) + \cdot \texttt{MI}(Z_{\texttt{mlp}}, Z_{\texttt{aug}})$
12     Gradient backward to update $\texttt{MLP}_\theta$ and $\texttt{FC}_\phi$
13 **Testing**
14 $\hat{Y} = \texttt{FC}_\phi[\texttt{MLP}_\theta(X)]$

---

# B PROOFS

## B.1 PROOF FOR THEOREM 1

**Theorem 1** *Let $\hat{A}$ be a function of $X$, i.e., $\hat{A} = f(X) \in \mathbb{R}^{N \times N}$, then $H(A|X)$ is upper bounded by $H(A|\hat{A})$. Formally:*

$$H(A|X) \le \inf_f H(A|\hat{A}). \tag{9}$$

*Proof.* Note that $\hat{A}$ is a function of $X$, as a result, there is $H(\hat{A}|X) = 0$. Then,

$$
\begin{aligned}
H(A|X) &= H(A) - I(A; X) \\
&= H(A) - (I(A; X; \hat{A}) + I(A; X|\hat{A})) \\
&= H(A) - (I(A; \hat{A}) - I(A; \hat{A}|X) + I(A; X|\hat{A})) \\
&= (H(A) - I(A; \hat{A})) - I(A; X|\hat{A}) \\
&= H(A|\hat{A}) - I(A; X|\hat{A}) \\
&\le H(A|\hat{A})
\end{aligned}
\tag{10}
$$

The equality can be attained if and only if $I(A; X|\hat{A}) = 0$. $\qquad\square$

## B.2 PROOF FOR THEOREM 2

**Theorem 2** *Assume the $\ell_2$ distance of positive (real-existing) edges follows a Gaussian distribution $p_{\texttt{pos}} \sim \mathcal{N}(\mu_p, \sigma_p^2)$, and the $\ell_2$ distance of negative (non-existing) edges follows another Gaussian*

distribution $p_{\text{neg}} \sim \mathcal{N}(\mu_n, \sigma_n^2)$. Let $\mu_m = \frac{1}{2}(\mu_p + \mu_n)$, and $\sigma_m^2 = \frac{1}{2}(\sigma_p^2 + \sigma_n^2)$. With the assumption that the probability that an edge is positive/negative is the same, the gap between $H(A|X)$ and $H(A|\hat{A})$ is

$$H(A|X) - H(A|\hat{A}) = I(A;X) - \mathcal{D}_{JS}(p_{\text{pos}}\|p_{\text{neg}}), \tag{11}$$

where $\mathcal{D}_{JS}$ denotes the Jensen-Shannon divergence, and

$$\mathcal{D}_{JS}(p_{\text{pos}}\|p_{\text{neg}}) = \frac{1}{2}\left(\frac{(\mu_p - \mu_m)^2 + (\mu_n - \mu_m)^2 + \sigma_p^2 + \sigma_n^2}{\sigma_m^2} - 2 + \log(\frac{\sigma_m^4}{\sigma_p^2 \sigma_n^2})\right) \tag{12}$$

*Proof.* The gap between $H(A|X)$ and $H(A|\hat{A})$ is computed as follows:

$$
\begin{aligned}
H(A|X) - H(A|\hat{A}) &= I(A;X|\hat{A}) \\
&= I(A;X) - I(A;X;\hat{A}) \\
&= \text{const} - I(A;\hat{A})
\end{aligned} \tag{13}
$$

Since $I(A;X)$ is a constant value not related to the formulation of $\hat{A}$, we only have to focus on $I(A;\hat{A})$, i.e., the mutual information between $A$ and $\hat{A}$. According to the definition of mutual information, we have:

$$
\begin{aligned}
I(A;\hat{A}) &= \sum_{A_{ij}} \sum_{\hat{A}_{ij}} p(A_{ij}, \hat{A}_{ij}) \log \frac{p(A_{ij}, \hat{A}_{ij})}{p(A_{ij})p(\hat{A}_{ij})} \\
&= \sum_{A_{ij}} \sum_{\hat{A}_{ij}} p(\hat{A}_{ij}|A_{ij}) p(A_{ij}) \log \frac{p(\hat{A}_{ij}|A_{ij})}{p(\hat{A}_{ij})}
\end{aligned} \tag{14}
$$

Since $\hat{A}_{ij}$ is the $\ell_2$ distance, we have $p(\hat{A}_{ij}|A_{ij} = 1) = p_{\text{pos}} \sim \mathcal{N}(\mu_p, \sigma_p)$, and $p(\hat{A}_{ij}|A_{ij} = 0) = p_{\text{neg}} \sim \mathcal{N}(\mu_n, \sigma_n)$. Furthermore, we can assume $p(A_{ij} = 0) = p(A_{ij} = 1) = 1/2$, then the above equation becomes:

$$
\begin{aligned}
I(A;\hat{A}) &= \frac{1}{2} \sum_{\hat{A}_{ij}} p_{\text{pos}}(\hat{A}_{ij}) \log \frac{p_{\text{pos}}(\hat{A}_{ij})}{\frac{1}{2}(p_{\text{pos}}(\hat{A}_{ij}) + p_{\text{neg}}(\hat{A}_{ij}))} \\
&\quad + \frac{1}{2} \sum_{\hat{A}_{ij}} p_{\text{neg}}(\hat{A}_{ij}) \log \frac{p_{\text{pos}}(\hat{A}_{ij})}{\frac{1}{2}(p_{\text{neg}}(\hat{A}_{ij}) + p_{\text{neg}}(\hat{A}_{ij}))} \\
&= \mathcal{D}_{JS}(p_{\text{pos}}\|p_{\text{neg}}),
\end{aligned} \tag{15}
$$

where $\mathcal{D}_{JS}$ is the JS-divergence. With the assumption that both $p_{\text{pos}}$ and $p_{\text{neg}}$ are Gaussian distributions, we have

$$I(A;\hat{A}) = \frac{1}{2}\left(\frac{(\mu_p - \mu_m)^2 + (\mu_n - \mu_m)^2}{\sigma_m^2} + \frac{\sigma_p^2 + \sigma_n^2}{\sigma_m^2} - 2 + \log\left(\frac{\sigma_m^4}{\sigma_p^2 \sigma_n^2}\right)\right) \tag{16}$$

Then, the proof is complete. □

### B.3 EQ. 7 AS MUTUAL INFORMATION MAXIMIZER

We adopt the following objective function (as shown in Eq. 7) to maximize the mutual information between $Z_{\text{mlp}}$ and $Z_{\text{aug}} - I(Z_{\text{mlp}}; Z_{\text{aug}})$:

$$\mathcal{L}_{\text{MI}} = -\alpha \sum_{i=1}^{N} \|\tilde{z}_i^{\text{mlp}} - \tilde{z}_i^{\text{aug}}\|_2^2 + \beta \sum_{p \neq q} C_{pq}^2,$$

where $z_i$ is the $i$-th row of $\tilde{Z}$, and $\tilde{Z}$ is the standardization of $Z$ along the instance dimension – $\tilde{Z} = Z - \mu(Z)/(\sigma(Z) \cdot \sqrt{N})$. $C \in \mathbb{R}^{D \times D}$ is the auto-correlation matrix of $Z_{\text{mlp}}$, and is computed as $C = \tilde{Z}^\top \tilde{Z}$.

Eq. 7 has been proved to be able to maximize the mutual information between $Z_{\mathtt{mlp}}$ and $Z_{\mathtt{aug}}$ in previous literature (Zhang et al., 2021a), below is a simplified version of the proof.

First, the mutual information between $Z_{\mathtt{mlp}}$ and $Z_{\mathtt{aug}}$ can be decomposed as follows:

$$I(Z_{\mathtt{mlp}}; Z_{\mathtt{aug}}) = H(Z_{\mathtt{mlp}}) - H(Z_{\mathtt{mlp}}|Z_{\mathtt{aug}}), \tag{17}$$

where $H(Z_{\mathtt{mlp}})$ is the entropy of $Z_{\mathtt{mlp}}$, $H(Z_{\mathtt{mlp}}|Z_{\mathtt{aug}})$ is the conditional entropy of $Z_{\mathtt{mlp}}$, given the observation of $Z_{\mathtt{aug}}$. We then seek to maximize $H(Z_{\mathtt{mlp}})$ and minimize $H(Z_{\mathtt{mlp}}|Z_{\mathtt{aug}})$, respectively.

To maximize $H(Z_{\mathtt{mlp}})$, assuming that $Z_{\mathtt{mlp}}$ follows a Gaussian distribution, then

$$\max H(Z_{\mathtt{mlp}}) \cong \max \log |C|, \tag{18}$$

where $|C|$ is the determinant of the covariance matrix of $Z_{\mathtt{mlp}}$. Since $Z_{\mathtt{mlp}}$ is column-standardized, $C$ is just the correlation matrix in Eq. 7. The diagonal entries of $C$ are all 1's. And $C \in \mathbb{R}^{D \times D}$ is a symmetric matrix. Let $\lambda_1, \lambda_2, \cdots, \lambda_D$ be the $D$ eigenvalues of $C$, then $\sum\limits_{i=1}^{D} \lambda_i = \mathrm{trace}(C) = D$. There is

$$\log |C| = \log \prod_{i=1}^{D} \lambda_i = \underbrace{\sum_{i=1}^{D} \log \lambda_i \leq D \log \frac{\sum\limits_{i=1}^{D} \lambda_i}{D}}_{\text{Jensen Inequality}} = 0. \tag{19}$$

This means that the upper bound of $|C|$ is 1, and the upper bound is achieved if and only if $\lambda_i = 1$ for $\forall i$, which indicates $C$ is an identity matrix and can be achieved by $\min \sum\limits_{p \neq q} C_{pq}^2$.

To minimize $H(Z_{\mathtt{mlp}}|Z_{\mathtt{aug}})$, according to the definition of conditional entropy:

$$\begin{aligned} H(Z_{\mathtt{mlp}}|Z_{\mathtt{aug}}) &= -\sum_{\boldsymbol{z}^{\mathtt{mlp}}} \sum_{\boldsymbol{z}^{\mathtt{aug}}} p(\boldsymbol{z}^{\mathtt{mlp}}, \boldsymbol{z}^{\mathtt{mlp}}) \log p(\boldsymbol{z}^{\mathtt{mlp}}|\boldsymbol{z}^{\mathtt{aug}}) \\ &= -\sum_{(\boldsymbol{z}_i^{\mathtt{mlp}}, \boldsymbol{z}_i^{\mathtt{aug}})} \log p(\boldsymbol{z}_i^{\mathtt{mlp}}|\boldsymbol{z}_i^{\mathtt{aug}}) \end{aligned} \tag{20}$$

Similarly, assuming $p(Z_{\mathtt{mlp}}|Z_{\mathtt{aug}})$ is a Gaussian distribution, then Eq. 20 becomes the MSE loss between $\boldsymbol{z}_i^{\mathtt{mlp}}$ and $\boldsymbol{z}_i^{\mathtt{aug}}$, which is the same as the first term in Eq. 7. Therefore the proof is complete.

## C  COMPUTING THE CONDITIONAL ENTROPY

In this section, we introduce how we estimate the conditional entropy $H(A|X)$ and $H(A|X_{\mathtt{aug}})$ presented in Fig. 2 and Fig. 3. Note that we estimate $H(A|X)$ / $H(A|X_{\mathtt{aug}})$ via a tractable upper bound $H(A|\hat{A})$, where $\hat{A}$ is the reconstructed adjacency matrix using $X$ or $X_{\mathtt{aug}}$. For simplicity in the following parts we take the raw node features $X$ as an example.

First, we compute $\hat{A}$ via:

$$\hat{A}_{ij} = \|\hat{\boldsymbol{x}}_i - \hat{\boldsymbol{x}}_j\|_2^2, \tag{21}$$

where $\hat{\boldsymbol{x}}_i$ and $\hat{\boldsymbol{x}}_j$ are $\ell_2$ normalized vectors of $\boldsymbol{x}_i$ and $\boldsymbol{x}_j$. The ground-truth of the adjacency information $A_{ij}$ is already known to us. Therefore, we wish to get conditional distribution $p(A_{ij}|\hat{A}_{ij})$, which could be achieved according to the Bayesian Rules:

$$\begin{aligned} p(A_{ij}|\hat{A}_{ij}) &= \frac{p(\hat{A}_{ij}|A_{ij})p(A_{ij})}{p(\hat{A}_{ij})} \\ &= \frac{p(\hat{A}_{ij}|A_{ij})p(A_{ij})}{p(\hat{A}_{ij}|A_{ij} = 0)p(A_{ij} = 0) + p(\hat{A}_{ij}|A_{ij} = 1)p(A_{ij} = 1)} \end{aligned} \tag{22}$$

Therefore,

$$p(A_{ij} = 1|\hat{A}_{ij}) = \frac{p(\hat{A}_{ij}|A_{ij} = 1)p(A_{ij} = 1)}{p(\hat{A}_{ij}|A_{ij} = 0)p(A_{ij} = 0) + p(\hat{A}_{ij}|A_{ij} = 1)p(A_{ij} = 1)} \tag{23}$$

and

$$p(A_{ij} = 0 | \hat{A}_{ij}) = \frac{p(\hat{A}_{ij} | A_{ij} = 0) p(A_{ij} = 0)}{p(\hat{A}_{ij} | A_{ij} = 0) p(A_{ij} = 0) + p(\hat{A}_{ij} | A_{ij} = 1) p(A_{ij} = 1)} \tag{24}$$

Eq. 23 and Eq. 24 are tractable due to the following reasons:

- $p(\hat{A}_{ij} | A_{ij} = 0)$ and $p(\hat{A}_{ij} | A_{ij} = 1)$ are tractable via density estimation.
- $p(A_{ij} = 0)$ and $p(A_{ij} = 1)$ can be computed directly by computing the density of graph.

However, the above estimation might be problematic as: 1) the graph can be very large, and it will be compute $p(A_{ij} | \hat{A}_{ij})$ for all $(i, j)$ pairs; 2) The real-world graphs are usually very sparse, indicting that the prior $p(A_{ij} = 0)$ can be close to 1, and $p(A_{ij} = 1)$ is close to 0. Such an imbalanced prior distribution might make the estimated conditional entropy unable to reflect the real divergence between the estimated graph structure and the real graph structure.

To mitigate this issue, we randomly sampled non-existing edges of the same size as the real existing edges, thus making $p(A_{ij} = 0) = p(A_{ij} = 1) = 1/2$. Then Eq. 23 and Eq. 24 become the following formulations:

$$p(A_{ij} = 1 | \hat{A}_{ij}) = \frac{p(\hat{A}_{ij} | A_{ij} = 1)}{p(\hat{A}_{ij} | A_{ij} = 0) + p(\hat{A}_{ij} | A_{ij} = 1)}$$
$$p(A_{ij} = 0 | \hat{A}_{ij}) = \frac{p(\hat{A}_{ij} | A_{ij} = 0)}{p(\hat{A}_{ij} | A_{ij} = 0) + p(\hat{A}_{ij} | A_{ij} = 1)} \tag{25}$$

If we denote the real-existing edge set by $\mathcal{E}^+$, while the sampled non-existing edge set by $\mathcal{E}^-$, then $H(A | \hat{A})$ could be computed by:

$$H(A | \hat{A}) = \frac{\sum\limits_{(i,j) \in \mathcal{E}^+} \log p(A_{ij} = 1 | \hat{A}_{ij}) + \sum\limits_{(i,j) \in \mathcal{E}^-} \log p(A_{ij} = 0 | \hat{A}_{ij})}{|\mathcal{E}^+| + |\mathcal{E}^-|} \tag{26}$$

Both $p(\hat{A}_{ij} | A_{ij} = 1)$ and $p(\hat{A}_{ij} | A_{ij} = 0)$ can be estimated using kernel density estimation by `sckit-learn`[1].

We present additional empirical results in Appendix E.8

## D  DETAILS OF EXPERIMENTS

### D.1  IMPLEMENTATIONS

We use pytorch to implement the proposed method. The Graph Convolution operation in Eq. 5 is implemented with DGL (Wang et al., 2019). We use Adam (Kingma & Ba, 2015) optimizer to train the model, and all the experiemnts are conducted on a NVIDIA RTX 4090 GPU with 24G memory.

### D.2  DATASETS

Following previous works (Zhang et al., 2021b; Hu et al., 2021), we consider 7 medium-sized graph datasets: Cora, Citeseer, Pubmed, Computer (McAuley et al., 2015), Photo, CS, and Physics (Sinha et al., 2015), as well as 4 large-scale graphs Flickr, Reddit, Yelp and Ogbn-Arxiv (Hu et al., 2020). We also include three heterophilic graphs Chameleon, Texas, and Cornel. We present the statistics of these datasets in Table 5. For both the transductive setting, inductive setting, and cold-start setting, we report the mean accuracy with standard deviation over 20 random trials.

The splits of these datasets are as follows: For Cora, Citeseer, and Pubmed, we directly use the public split, where 20 nodes are for training, 1500 nodes are for validation, and 1000 nodes are for

---

[1]https://scikit-learn.org/stable/modules/generated/sklearn.neighbors.KernelDensity.html

testing. For `Computer`, `Photo`, `CS`, and `Physics`, no splits are provided, so we randomly split the nodes into training/validation/testing with $8:1:1$ ratio. For `Flickr`, `Reddit` and `Yelp`, we use the split provided by the paper GraphSAINT (Zeng et al., 2020). The splits of heterophilic graphs are from Pei et al. (2020).

Table 5: Statistics of benchmark datasets. Yelp has multiple labels.

| Dataset | #Nodes | #Edges | #Classes | #Features | Train/Val/Test |
|---|---|---|---|---|---|
| Cora | 2,708 | 10,556 | 7 | 1,433 | 140/1500/1000 |
| Citeseer | 3,327 | 9,228 | 6 | 3,703 | 120/1500/1000 |
| Pubmed | 19,717 | 88,651 | 3 | 500 | 60/1500/1000 |
| CS | 18,333 | 327,576 | 15 | 6,805 | 80%/10%/10% |
| Physics | 34,493 | 991,848 | 5 | 8,451 | 80%/10%/10% |
| Computer | 13,752 | 574,418 | 10 | 767 | 80%/10%/10% |
| Photo | 7,650 | 287,326 | 8 | 745 | 80%/10%/10% |
| Chameleon | 2,277 | 31,371 | 5 | 2,325 | 60%/20%/20% |
| Texas | 183 | 279 | 5 | 1,703 | 60%/20%/20% |
| Cornell | 183 | 277 | 5 | 1,703 | 60%/20%/20% |
| Flickr | 89,250 | 899,756 | 7 | 500 | 0.50/0.25/0.25 |
| Reddit | 232,965 | 114,615,892 | 41 | 602 | 0.66/0.10/0.24 |
| Arxiv | 169,343 | 2,332,486 | 40 | 128 | 90,941/ 29,799/48,603 |
| Yelp | 716,847 | 6,977,410 | 100 | 300 | 0.75/0.10/0.15 |

### D.3 HYPER-PARAMETERS.

We select the optimal hyperparameter setting for each dataset via a small grid search. The optimal hyperparameter settings have been presented in the Supplementary, and here we present the range that we perform grid search:

- Learning rate: 1e-3
- Weight decay: [1e-4, 1e-5, 1e-6]
- Hidden dimension: [256, 512, 1024]
- Num layers of MLP: 2
- dropout: [0.1, 0.2, 0.3]
- $\alpha$: [5e-4, 1e-3, 2e-3]
- $\beta$: [5e-7, 1e-6, 5e-6]

We also tune the hyperparameters of baseline methods for fair comparison. Specifically, Graph Neural Networks usually perform well with low hidden dimensions, therefore we tune the hidden dimensions of GNNs within the range [32, 64, 128, 256, 512, 1024]. Besides, since the number of propagation $K$ plays the same role of the number of GNN layers, we tune the number of layers for GNN baselines from 1 to 10. We report the best performance in Table 3 and Table 4.

### D.4 DISCUSSION OF THE INFORMATION LEAKAGE IN KNOWLEDGE DISTILLATION MLPs

Although knowledge distillation methods have been shown effective for learning student MLPs on graphs via distilling the predictions from teacher GNNs on a variety of settings (Zhang et al., 2021b; Tian et al., 2022), they mistakenly use the teacher GNN's predictions (logits) of validation and testing nodes to guide the student MLP, enforcing the testing nodes of student MLP model to mimic the predictions of the teacher GNN model directly. A direct consequence of this practice is that, as long as the student MLP model has enough parameters, it can always obtain the same predictions as the teacher GNN. This issue was observed by the official reviewer of GLNN paper[2]: "the MLP in the transductive setting is trained to reproduce the logits of the GNN on all nodes of the given graph, regardless of whether these belong to the training, validation, or test set. If the nodes thus all show

---

[2]https://openreview.net/forum?id=4p6_5HBWPCw

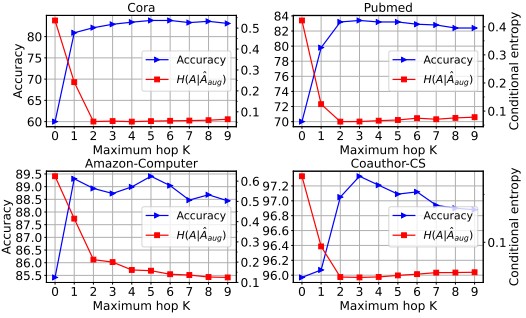

Table 6: Comparison of training/testing time and accuracy of InfoMLP with the baseline models, on `Pubmed` dataset, under the transductive setting. **InfoMLP** has an additional pre-processing step that takes 0.0025s.

Figure 3: The change of test accuracy (the blue curve), and the conditional entropy $H(A|\hat{A}_{\text{aug}})$ (an upper bound of $H(A|X_{\text{aug}})$, the red curve), with respect to $K$. $K = 0$ corresponds to the original node features, i.e., $X_{\text{aug}} = X$.

| Dataset | Training | Testing | Accuracy |
|---|---|---|---|
| SGC | 5.1689s | 0.0006s | 78.9 |
| GCN | 19.8859s | 0.0086s | 79.3 |
| GAT | 53.9234s | 0.0117s | 79.0 |
| JKNET | 38.0755s | 0.0060s | 78.9 |
| APPNP | 20.7649s | 0.0047s | 80.3 |
| Vanilla MLP | **3.0136**s | **0.0002**s | 68.4 |
| GraphMLP | 88.1919s | **0.0002**s | 79.7 |
| N2N | 74.4125s | **0.0002**s | 80.7 |
| GLNN | 28.3841s | **0.0002**s | 80.5 |
| **InfoMLP** | 3.5999s | **0.0002**s | **83.2** |

different features, a sufficiently large and deep MLP should always be able to overfit on the features of the target nodes and replicate the predictions of the GNN."

Consequently, as acknowledged by the authors of GLNN (Zhang et al., 2021b), the transductive setting of GLNN can only serve as a sanity check, showing that MLPs have the capacity to reproduce the predictions of GNNs using knowledge distillation. The subsequent knowledge distillation work, NOSMOG (Tian et al., 2022) follows the setting of GLNN. Therefore, the information leakage issue still exists. To fix this issue, in our reproduction of GLNN and NOSMOG, we exclude validation and testing nodes from the node-set where knowledge distillation is applied.

# E    ADDITIONAL EMPIRICAL RESULTS

## E.1    FURTHER STUDY OF INFOMLP

**Impacts of the maximum propagation steps $K$.**    In Section 3.3 and Equation 5, we described the minimization of $H(A|X_{\text{aug}})$ as a preprocessing step, where the hyperparameter $K$ controls the maximum propagation step. In order to empirically examine the relationship between InfoMLP's performance and the conditional entropy $H(A|X_{\text{aug}})$ and $K$, we estimate the value of $H(A|X_{\text{aug}})$ (approximated with $H(A|\hat{A}_{\text{aug}})$) with different values of $K$. We then plot the accuracy and entropy curves with respect to $K$ in Figure 3. Due to space limitations, we only present the results for the `Cora`, `Pubmed`, `Computer`, and `CS` datasets here. The results for the remaining datasets, as well as the positive/negative distributions of all datasets, can be found in Appendix D. In Figure 3, we observe a clear negative correlation between the conditional entropy and performance, which confirms the rationale behind our design choices. We also observe that the correlations are stronger for `Cora` and `Pubmed` compared to `Computer` and `CS`, which is consistent with the results in Table 2 and Figure 2. On `Computer`, where the correlation between features and structure is minimal, and on `CS`, where the correlation is significant, the propagation of features over the graph structure is less influential in achieving high performance. These observations further support the effectiveness and rationality of our design choices in InfoMLP.

**Comparison of training/testing time.**    In Table 6, we compare the training and testing times of InfoMLP with other models on the `Pubmed` dataset. The training time represents the total time for training over 500 epochs, while the testing time is for a single evaluation. We observe that the training time of InfoMLP is only slightly slower than the vanilla MLP, but significantly faster than GNNs and other MLP models. Despite the slight increase in training time, InfoMLP achieves superior performance on the node classification task. This highlights the efficiency and effectiveness of our proposed method. Additional results for large-scale datasets can be found in Appendix E.

### E.2 COMPARISON WITH ADVANCED BASELINES OF MULTI-HOP INFORMATION

In this section, we compare InfoMLP with another two advanced GNN models that consider higher-order information: GPR-GNN (Chien et al., 2021) and PPGNN Lingam et al. (2022). The results are presented in Table 7. Overall, they achieve similar performance to InfoMLP on Cora, but clearly worse performance on others.

Table 7: Performance comparison of InfoMLP with GPR-GNN and PPGNN on the public split of citation netwroks. GPR-GNN and PPGNN can achieve close performance on Cora, while clearly worse performance on Citeseer and Pubmed, than the proposed InfoMLP.

| Dataset | Cora | Citeseer | Pubmed |
|---------|------|----------|--------|
| GPR-GNN | 83.4 | 72.0 | 79.0 |
| PPGNN | 83.5 | 71.5 | 79.3 |
| InfoMLP | **83.8** | **73.7** | **83.2** |

### E.3 RESULTS WITH OTHER LABELING RATES

The citation networks Cora, Citeseer, and Pubmed use the public split, where 20 nodes per class are used as the training nodes. We would like to investigate the performance change of different methods with different labeling rates. Therefore, we create four training sets of $10/20/40/80$ nodes per class, and then compare the performance of GCN (Kipf & Welling, 2017), GPR-GNN Chien et al. (2021), PPGNN (Lingam et al., 2022), and our InfoMLP in Table 8. The results show that the proposed InfoMLP significantly outperforms GNN baselines when the labeling rate is low, e.g., 10 or 20 training nodes per class. The gaps between GNNs (especially GPR-GNN) and InfoMLP narrow as the labeling rate grows, while the performance of InfoMLP is still very competitive even when the label rate increases. Since lower label rates are more challenging and more common in real-world scenarios, The outstanding performance of InfoMLP at low labeling rates further highlights its value.

Table 8: The performance of all methods gets improved as the label rate increases.

| Dataset | Cora | | | | Citeseer | | | | Pubmed | | | |
|---------|------|------|------|------|----------|------|------|------|--------|------|------|------|
| # Nodes per class | 10 | 20 | 40 | 80 | 10 | 20 | 40 | 80 | 10 | 20 | 40 | 80 |
| GCN | 75.8 | 81.9 | 82.8 | 84.5 | 66.9 | 71.6 | 72.9 | 74.1 | 73.2 | 79.3 | 80.7 | 82.4 |
| GPR-GNN | 76.9 | 83.4 | 84.4 | **86.2** | 67.1 | 72.0 | 73.8 | 74.9 | 74.2 | 79.0 | 80.7 | 83.2 |
| PPGNN | 76.6 | 83.5 | 84.3 | 85.4 | 67.7 | 71.5 | 73.1 | 73.9 | 74.9 | 79.3 | 81.1 | 83.4 |
| InfoMLP | **81.2** | **83.8** | **84.6** | 85.8 | **70.8** | **73.7** | **74.3** | **75.1** | **80.4** | **83.2** | **84.1** | **85.4** |

### E.4 APPLICATION TO HETEROPHILIC/FEATURE-CENTRIC GRAPHS

In this section, we study if the proposed InfoMLP can also applied to heterophilic graphs. In Table 9, we compare the performance of InfoMLP with the two aforementioned GNN methods (which have been shown to perform well on heterophilic graphs), on three representative heterophilic graphs – Chameleon, Texas and, Cornell. Note that in Table 9, the graph-augmented node feature matrix is obtained by Eq. 27.

$$X_{\mathrm{aug}}(K) = g(X, A) = \sum_{k=1}^{K} \gamma_k \tilde{A}^k X, \ \gamma_k = 1/K. \tag{27}$$

As shown in Table 9, the proposed InfoMLP is able to obtain comparable performance of two feature-centric graphs, Texas and Cornell, while failing to obtain satisfying performance on Chameleon. The following two tricks can be applied for adaptation to the heterophilic/feature-centric graphs.

On feature-centric graphs where graph structure information might be useless or even harmful, we may simply set $\alpha, \beta = 0$ such that InfoMLP is equivalent to the vanilla MLP, guaranteeing it performs no worse than MLP.

Table 9: InfoMLP with the vanilla design of node-augmented feature matrix does perform well on heterophily graph Chameleon, but can achieve comparable performance on feature-centric datasets Texas and Cornell.

| Dataset | Chameleon | Texas | Cornell |
|---------|-----------|-------|---------|
| GPR-GNN | 62.59 | 81.35 | 78.11 |
| PPGNN | **67.74** | **89.73** | 82.43 |
| InfoMLP | 54.16 | 80.42 | **82.72** |

For handling heterophily, we can flexibly combine InfoMLP with other techniques. For example, we can adjust the weight coefficients $\gamma_k$ in Eq. 5 and Eq. 27, such that it assigns different weights to different hops of information. For verification, we apply the weights $\gamma_k$ for $k = 1$ to 10 from GPR-GNN model (Chien et al., 2021) on Chameleon. The learned weights are presented in Table 10. Consequently, as shown by the results in Table 11, the performance of InfoMLP increases from 54.16 to 61.98 on Chameleon, approaching that of GPR-GNN.

Table 10: Weights of $\gamma_i, i = 1, \cdots, 10$ from GPR-GNN, which are applied to the refined graph-augmented node feature matrix $X_{\mathrm{aug}}(K) = \sum\limits_{k=1}^{K} \gamma_k \tilde{A}^k X$.

| $\gamma_1$ | $\gamma_2$ | $\gamma_3$ | $\gamma_4$ | $\gamma_5$ | $\gamma_6$ | $\gamma_7$ | $\gamma_8$ | $\gamma_9$ | $\gamma_{10}$ |
|-----------|-----------|-----------|-----------|-----------|-----------|-----------|-----------|-----------|-----------|
| -0.8757 | 2.8925 | 1.2931 | 0.4811 | -0.1479 | -0.4435 | -0.5420 | -0.5979 | -0.5579 | -0.5374 |

Table 11: Performance comparison between GPR-GNN, the vanilla InfoMLP, and revised InfoMLP with graph-augmented node feature matrix of adjustable weights. The vanilla InfoMLP with fixed weights does not perform well on GPR-GNN, while the revised one achieves comparable performance than GPR-GNN.

| Dataset | Chameleon |
|---------|-----------|
| GPR-GNN | 62.59 |
| InfoMLP (vanilla Eq.4) | 54.16 |
| InfoMLP (revised Eq.4) | 61.98 |

Furthermore, we plot the distribution figure of the original node features $X$, the vanilla graph-augmented feature (with fixed $\gamma_k = 1/K$), and the revised graph-augmented feature (with $\gamma_k$ in Table 10) in Fig. 4. As observed in Fig. 4, the positive edges and negative edges can hardly be discriminated by the original node features $X$. The graph-augmented node feature matrix $X_{\mathrm{aug}}(K)$ of the vanilla definition can better discriminate positive edges and negative edges, while the conditional entropy $H(A|\hat{A})$ is still large. The revised definition of $X_{\mathrm{aug}}(K)$ with adjustable weights can better discriminate positive/negative edges. Therefore, $H(A|\hat{A})$ is the smallest. The results in this figure align with the numerical results in Table 11, therefore verifying our analysis.

### E.5 EXTENDING MLPs TO LARGE-SCALE GRAPHS

In Table 12, we first compare the performance of InfoMLP with GNN and MLP baselines on four large-size graphs – Flickr, Reddit, Yelp, and Arxiv. It is worth noting that the two knowledge distillation methods GLNN (Zhang et al., 2021b) and NOSMOG ()nosmog fail to give a competitive performance after fixing the information leakage issue as claimed on Arxiv. As shown in Table 12, the vanilla MLP usually performs much poorer than GNNs on large graphs, as they fail to utilize the rich graph structure information for prediction. It is The advanced MLP methods' performance can be improved to be close to GNNs' on Flickr and Yelp datasets. However, on Reddit and Arxiv, the performance gap between MLPs and GNNs remains substantial, even when using advanced MLP models. We speculate that the reasons for this may be as follows.

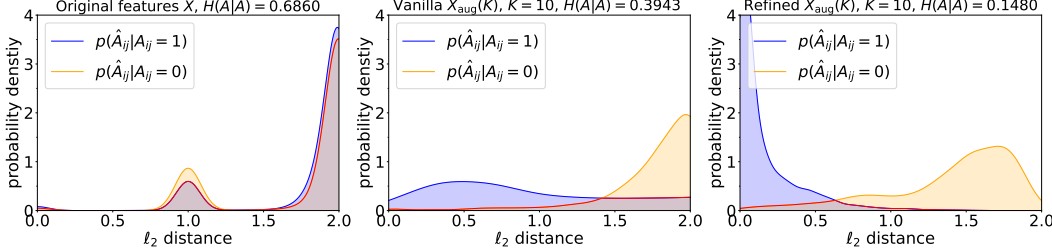

Figure 4: Estimated probability density function of $p(\hat{A}|A)$ and conditional entropy $H(A|\hat{A})$ of `Chameleon` dataset. Positive edges and negative edges can hardly be discriminated by the original node features $X$. The graph-augmented node feature matrix $X_{\text{aug}}(K)$ of the vanilla definition can better discriminate positive edges and negative edges, while the conditional entropy $H(A|\hat{A})$ is still large. The revised definition of $X_{\text{aug}}(K)$ can better discriminate positive/negative edges. Therefore, $H(A|\hat{A})$ is smaller.

Table 12: Performance on large-scale graphs in the transductive setting. We use the micro-F1 score as the metric.

| Dataset | **Flickr** | **Reddit** | **Yelp** | **Arxiv** |
|---|---|---|---|---|
| SGC | 49.8±0.5 | 90.1±0.2 | 25.8±0.3 | 70.59±0.33 |
| GCN | 50.2±0.3 | 93.3±0.1 | 28.1±0.5 | 71.23±0.15 |
| Vanilla MLP | 46.5±0.5 | 52.8±0.3 | 22.3±0.8 | 56.03±0.21 |
| GraphMLP | 47.2±0.4 | 58.1±1.2 | 24.5±0.5 | 56.72±0.26 |
| GLNN | 47.6±0.3 | 65.7±0.9 | 26.3±0.4 | 57.19±0.36[1] |
| NOSMOG | 47.8±0.3 | 68.4±0.8 | 26.9±0.5 | 57.83±0.18[1] |
| InfoMLP | 48.5±0.4 | 71.4±0.3 | 27.1±0.6 | 58.64±0.35 |

[1] The results of GLNN and NOSMOG are inconsistent with the reported ones, because we've fixed the information leakage issues. See discussions in Appendix D.4

First, the performance of the vanilla MLP is very close to GNNs on `Flickr` and `Yelp`, indicating that the node features of the two datasets are really powerful for training a classifier. In this scenario, training a structure-regularized MLP model can help the model leverage additional structural information, allowing MLP to achieve similar performance to GNNs.

For the `Reddit` and `Arxiv` datasets, we observe a significant performance gap between vanilla MLPs and GNN methods, indicating that their node features alone are not powerful enough, and the graph structure provides more useful information for classification. In this scenario, we need to explore how to use advanced MLP methods to exploit this structural information.

Our analysis in Sec. 3.1 has revealed the mutual information between node features $X$ and the graph structure $A$: $I(X;A)$ directly constrains the capacity of MLPs. We apply our quantification method proposed in Sec. 3.2 to the two datasets, and the corresponding distributions are presented in Fig. 5. From Fig. 5, we can observe that for both `Arxiv` and `Reddit`, the distribution of positive edges and negative edges are heavily overlapped, indicating that it is difficult to discriminate positive edges from negative edges using the score function defined in Sec. 3.2, i.e., $\hat{A}_{ij} = \ell_2^2(\hat{x}_i, \hat{x}_j) = \|\hat{x}_i - \hat{x}_j\|_2^2$. Therefore $H(A|\hat{A})$ is large, and $I(X;A)$ is small. Consequently, the performance of MLPs can hardly be improved to the same level as GNNs, given the limited information overlapping between node features and the graph structure.

Typically, the node features $X$ is an $N \times d$ matrix, whereas the graph structure can be expressed as a sparse matrix of $E$ entries, where $N$ is the number of nodes, $E$ is the number of edges, $d$ is the dimension of node feature. Therefore, for graphs where $E$ is very large, the node feature matrix $X$ cannot express the graph structure $A$, and $I(X;A)$ is very small. The above intuition indicates that the density of the graph, rather than the number of nodes, affects the performance gaps between MLPs and GNNs. To be detailed, the structure of a sparse graph is easier to learn compared with a

Table 13: Performance on large-scale graphs in the inductive setting. We use the micro-F1 score as the metric.

| Method | **Reddit** | **Arxiv** |
|--------|-----------|-----------|
| GCN | $59.7 \pm 1.2$ | $56.7 \pm 0.5$ |
| InfoMLP | $66.3 \pm 1.4$ | $58.2 \pm 0.4$ |

Table 14: Forward time of one epoch on large-scale graphs.

| Dataset | **Flickr** | **Reddit** |
|---------|-----------|-----------|
| MLP | 30ms | 34.9ms |
| SGC | 350ms | 850ms |
| GCN | 4000ms | 22950ms |
| InfoMLP | 103.9ms | 310ms |

dense graph for the MLPs. To verify this, we conduct ablation experiments on `Arxiv` dataset. We subsample the edges of the graph and study the impacts on the conditional entropy $H(A|\hat{A})$.

Instead of the previously defined non-parametric formulation $\hat{A}_{ij} = \ell_2^2(\hat{\boldsymbol{x}}_i, \hat{\boldsymbol{x}}_j) = \|\hat{\boldsymbol{x}}_i - \hat{\boldsymbol{x}}_j\|_2^2$, we have to use a parameterized entropy estimator for more accurate estimation. We use a two-layer MLP to learn $\hat{A}_{ij}$:

$$\hat{A}_{ij} = \texttt{FC}(\texttt{ReLU}(\texttt{FC}(\texttt{CONCAT}(\boldsymbol{x}_i, \boldsymbol{x}_j)))) \tag{28}$$

The MLP's hidden dimension is 32, and the output dimension is 1. The MLP's parameters are optimized by a cross-entropy loss function between positive edges and negative edges. We apply the novel formulation of $\hat{A}_{ij}$ to compute the distributions of positive edges and negative edges, as well as the conditional entropy $H(A|\hat{A})$. Results in Fig. 6 demonstrate that as the number of edges increases (the graph becomes more dense), the positive edges and negative edges become harder to discriminate, and the conditional entropy becomes larger, indicating a smaller $I(X; A)$.

We further compare the performance of GCN and the proposed InfoMLP on the subsampled `Arxiv` dataset in Table 15. The change of the performance gap (relative error) is also plotted in Fig. 7. As clearly presented, as the number of subsampled edges decreases, the performance of InfoMLP quickly approaches GCN's. These above results justify our speculation that the huge gap between the information volume of node features and the rich graph information brought by a dense graph has caused the performance gap between GNNs and MLPs.

For most graph data, raw textual features might contain ample information to infer the structure of the graph. However, conventional techniques like skip-gram, word2vec, or GloVe, as used in `Reddit` and `Arxiv`, lead to a substantial loss of information. In Reddit's case, the inputs are low-dimensional features processed through GloVe, with the dataset encompassing over 230k nodes and more than 114m edges. In `Arxiv`, the input feature of each node (a paper) is a 128-dimensional feature vector obtained by averaging the embeddings of words (obtained via skip-gram) in its title and abstract. The low-quality feature extractors create a huge information gap between the node feature matrix and the graph structure,

To extend MLPs to larger and denser graphs, we have to reduce the information gap between the node features and the graph structure. One potential solution to shorten the information gap between node features and graph structure is to employ advanced feature extractors such as Large Language Models (LLM). With impressive capabilities in understanding, representing, and generating natural languages, the LLM can derive more expressive, high-quality node features from raw texts, thereby enabling the model to discover complex structural information from pure node features.

### E.6 COMPARISON WITH $Z_{\text{AUG}}$ AND $Z_{\text{MLP}}$

$Z_{\text{aug}}$ cannot be directly compared to $Z_{\text{mlp}}$ because in our model, the cross-entropy loss for downstream classification tasks is applied to $Z_{\text{mlp}}$ rather than $Z_{\text{aug}}$. To meet the reviewer's request, we

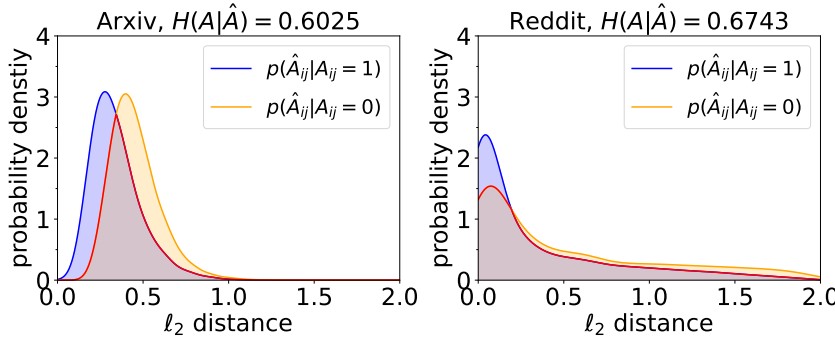

Figure 5: Estimated probability density function of $p(\hat{A}|A)$ of `Arxiv` and `Reddit` dataset. Blue line/curve stands for real edges ($A_{ij} = 1$) while orange line/curve stands for non-existing ones ($A_{ij} = 0$).

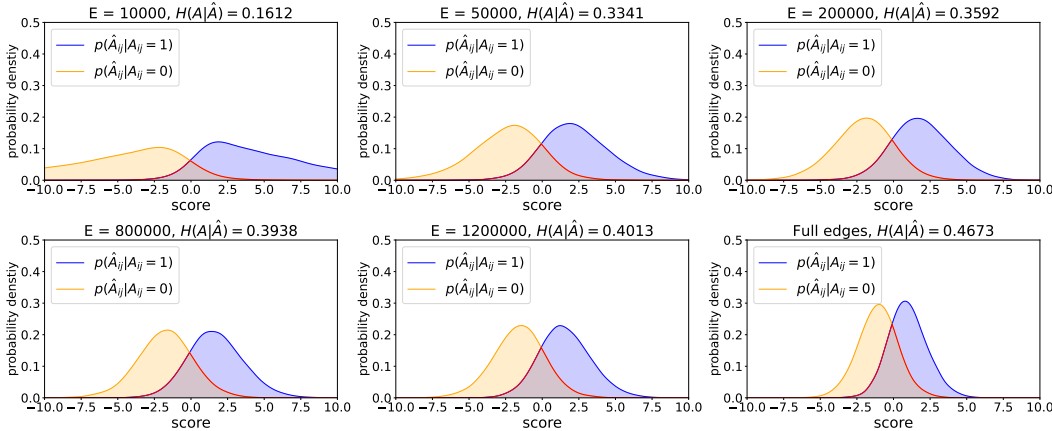

Figure 6: Distributions of positive edges and negative edges, and conditional entropy of `Arxiv` with different numbers of subsampled edges. E is the number of subsampled edges.

independently trained a classification model based on $Z_{\mathrm{aug}}$. In the table below, we provide the performance obtained with $Z_{\mathrm{aug}}$ in the transductive setting and compare it to $Z_{\mathrm{mlp}}$.

Table 16: Performance comparison between $Z_{\mathrm{aug}}$ and $Z_{\mathtt{mlp}}$.

| Embedding | **Cora** | **Citeseer** | **Pubmed** |
|---|---|---|---|
| $Z_{\mathrm{aug}}$ | $81.9 \pm 0.5$ | $72.5 \pm 0.5$ | $80.2 \pm 0.4$ |
| $Z_{\mathrm{mlp}}$ | $83.8 \pm 0.3$ | $73.7 \pm 0.3$ | $83.2 \pm 0.9$ |

As we can see, $Z_{\mathrm{aug}}$ generally performs worse on these datasets compared to $Z_{\mathrm{mlp}}$. We suspect that this is because the regularization loss employed by InfoMLP not only helps $Z_{\mathrm{mlp}}$ learn graph structural information but also includes a decorrelation loss, which encourages different dimensions of the embedding to be relatively independent. This, in turn, helps $Z_{\mathrm{mlp}}$ to obtain an embedding space that is more distinguishable and beneficial for classification.

Furthermore, since $Z_{\mathrm{aug}}$ itself utilizes graph structural information as an input to the encoder, it is natural that it can achieve good performance when node features alone do not contain sufficient graph structural information. For example, on the Arxiv dataset, $Z_{\mathrm{aug}}$ effortlessly achieved a classification accuracy of $70.82 \pm 0.16$.

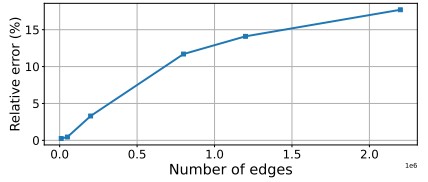

Figure 7: The performance gaps between GCN and the proposed InfoMLP with subsampled edges on `Arxiv` dataset.

Table 15: Test accuracy on subsampled `Arxiv`.

| # Edges  | 10k   | 50k   | 200k  | 800k  | 1.2m  | Full  |
|----------|-------|-------|-------|-------|-------|-------|
| GCN      | 55.87 | 56.69 | 58.94 | 65.03 | 67.52 | 71.23 |
| InfoMLP  | 56.01 | 56.43 | 56.99 | 57.43 | 57.99 | 58.64 |
| Error(%) | 0.25% | 0.45% | 3.3%  | 11.7% | 14.1% | 17.7% |

### E.7 PERFORMANCE USING OTHER MI ESTIMATORS

In addition to the loss used in this paper, we supplement new experiments with the InfoNCE MI estimator (van den Oord et al., 2018) and the Mutual Information Neural Estimator (MINE) (Belghazi et al., 2018). They have both been proven to have lower bounds on MI, making it suitable as the learning objective (the MI can maximized by maximizing the estimator). In the following table, we present the mean accuracy of InfoMLP with these estimators:

Table 17: Performance comparison by trying other MI estimators.

| MI Estimator | Cora | Citeseer | Pubmed |
|--------------|------|----------|--------|
| MINE (Belghazi et al., 2018) | $82.7 \pm 0.9$ | $71.9 \pm 0.8$ | $80.8 \pm 0.9$ |
| InfoNCE (van den Oord et al., 2018) | $84.0 \pm 0.4$ | $73.4 \pm 0.4$ | $82.9 \pm 0.3$ |

The InfoNCE MI estimator provides competitive performance as the feature-decorrelation-based loss used in our paper (even better on some datasets), while MINE in general, yields inferior performance. This is because MINE has a very large variance in MI estimation, which results in less accurate estimates. Although InfoNCE provides strong results, its computational complexity is $O(N^2 d)$, which makes it time and memory-consuming to calculate the InfoNCE loss on medium-sized graphs. Considering these, we use the estimator, which is efficient and produces competitive results in our paper.

### E.8 ADDITIONAL RESULTS ABOUT $H(A|X)$ AND $H(A|X_{\text{AUG}})$

In Sec. 3.1, we only present the figures demonstrating the overlapping information between $A$ and $X$ for five datasets due to the space limit. Here we provide the complete figures of all seven datasets in Fig. 8.

We further plot the distributions for positive/negative edges of `Cora`, `Citeseer`, `Pubmed`, `Computer`, `Photo`, and `CS` using the augmented node features $X_{\text{aug}}$ to compute $\hat{A}$. From the figures, we can obverse that increasing the maximum propagation step $K$ to get $X_{\text{aug}}$ helps discriminate the positive edges and negative edges using the augmented node features of `Cora`, `Citeseer`, `Pubmed`, and `CS`. However, for `Computer` and `Photo` (shown in Fig. 12 and Fig. 13), we can observe that the two distributions become even closer and thus harder to discriminate. In this case, the graph augmented node feature matrix contains less information about the graph structure, so structural regularization does not take effect.

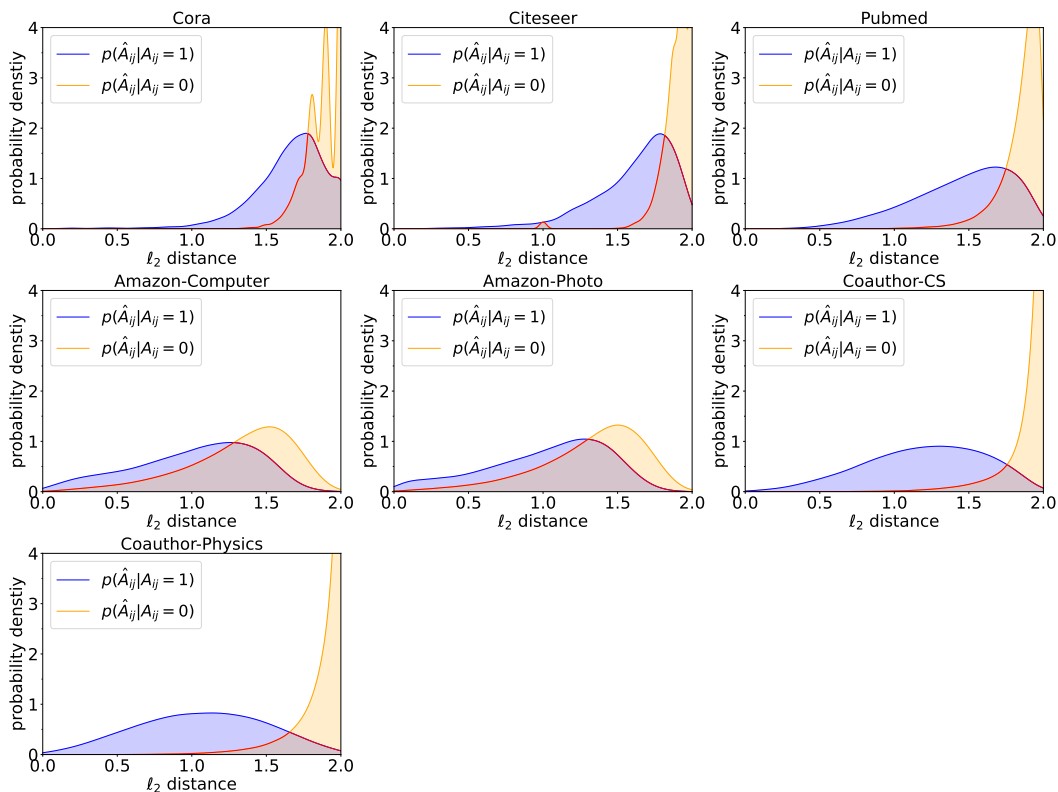

Figure 8: Estimated probability density function of $p(\hat{A}|A)$. Blue line/curve stands for real edges ($A_{ij} = 1$) while orange line/curve stands for non-existing ones ($A_{ij} = 0$).

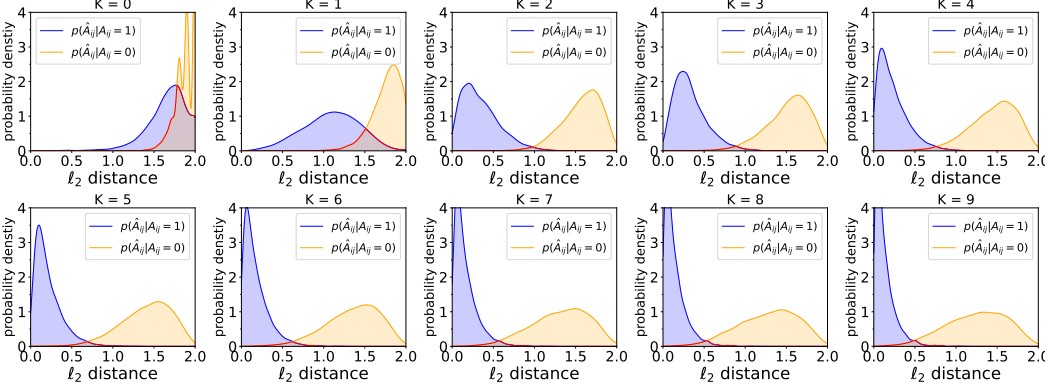

Figure 9: Cora: Overlapping between $A$ and $X_{\mathtt{aug}}$ according to the original adjacency matrix $A$ and the estimated one $\hat{A}$., with respect to the maximum propagation step $K$.

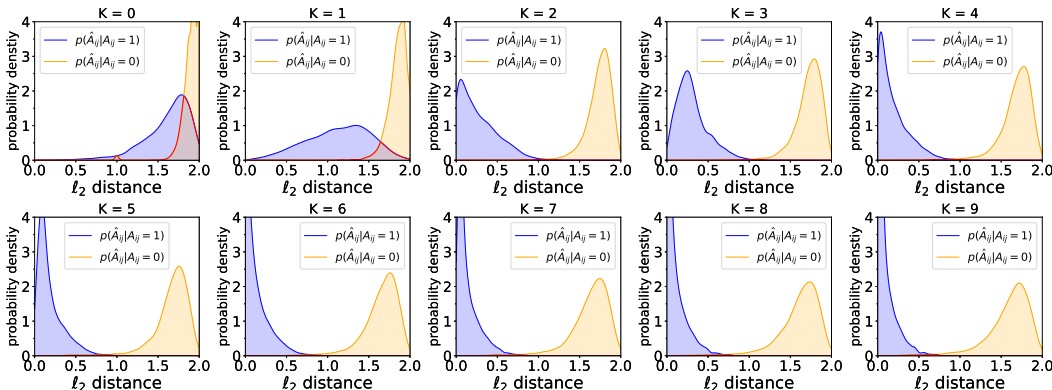

Figure 10: Citeseer: Overlapping between $A$ and $X_{\mathtt{aug}}$ according to the original adjacency matrix $A$ and the estimated one $\hat{A}$., with respect to the maximum propagation step $K$.

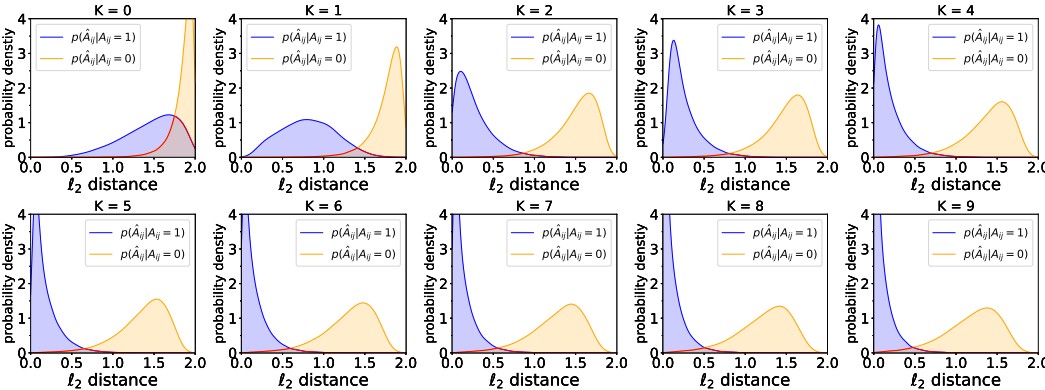

Figure 11: Pubmed: Overlapping between $A$ and $X_{\mathtt{aug}}$ according to the original adjacency matrix $A$ and the estimated one $\hat{A}$., with respect to the maximum propagation step $K$.

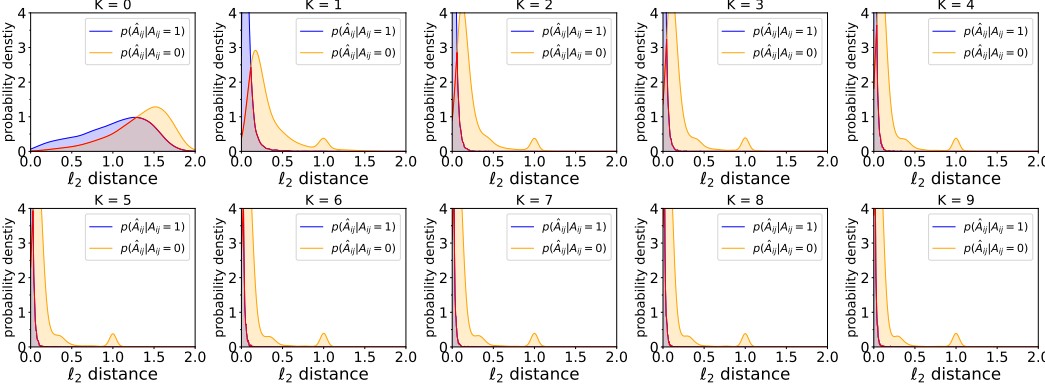

Figure 12: Amazon-Computer: Overlapping between $A$ and $X_{\mathtt{aug}}$ according to the original adjacency matrix $A$ and the estimated one $\hat{A}$., with respect to the maximum propagation step $K$.

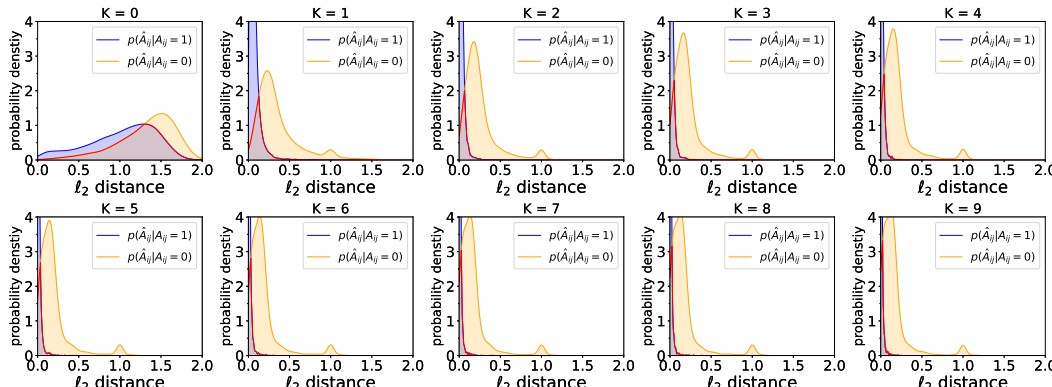

Figure 13: Amazon-Photo: Overlapping between $A$ and $X_{\texttt{aug}}$ according to the original adjacency matrix $A$ and the estimated one $\hat{A}$., with respect to the maximum propagation step $K$.

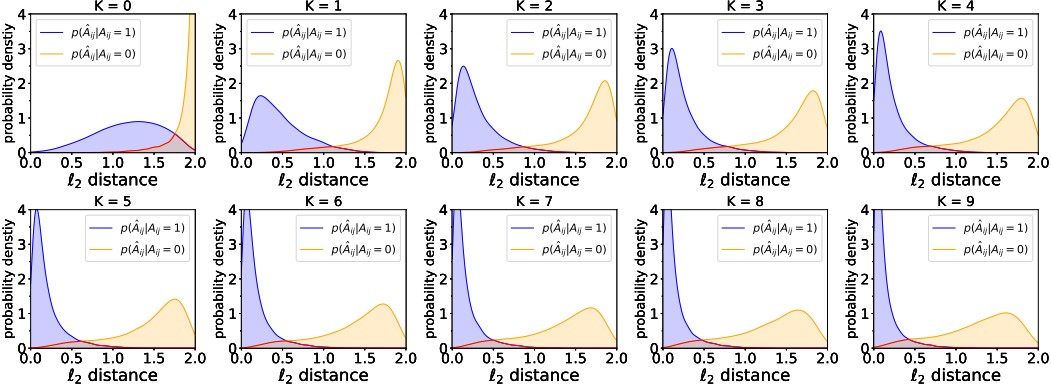

Figure 14: CS: Overlapping between $A$ and $X_{\texttt{aug}}$ according to the original adjacency matrix $A$ and the estimated one $\hat{A}$., with respect to the maximum propagation step $K$.

