# OpenReview forum: "When Do MLPs Excel in Node Classification? An Information-Theoretic Perspective"
_ICLR.cc/2024/Conference — Submitted to ICLR 2024_

### Official Review · Reviewer_zabj · 2023-10-25

**Soundness:** 1 poor
**Presentation:** 2 fair
**Contribution:** 2 fair
**Rating:** 3
**Confidence:** 4

**Summary:**

This paper studies the problem of node classification given node features and the graph structure information. The author propose a crude estimate on the extent to which node features cover graph structure information. Based on the idea of maximizing the mutual information between node embeddings and the graph structure, the authors introduce a novel regularization-based MLP model termed InfoMLP. InfoMLP consists of a preprocessing step and a learning step. Only the preprocessing step requires utilizing the graph structure information. Once the raw node features are preprocessed, InfoMLP has the same efficiency as a standard MLP during both the training and testing phases. Empirically, the proposed method demonstrates competitive performance over a few small benchmark datasets, while being much more efficient than existing graph neural networks.

**Strengths:**

- Efficiency is a major problem for the application of conventional GNNs in the industry, where practitioners often have to deal with graphs of massive size. This paper proposes an alternative and efficient method for node classification while utilizing both the raw node features and the graph structure.

- The idea to construct node embeddings during a preprocessing stage by explicitly maximizing the mutual information between node embeddings and the graph structure information is interesting, although not surprising. I liked this perspective.

- The empirical results over the selected benchmark datasets are promising, as shown throughout Section 4 in the paper.  (The additional results in the appendix, however, are on the pessimistic side, more on this later.)

**Weaknesses:**

- A number of claims in Section 3.1 are not well justified by either theoretical or empirical evidence. Here are some examples:
  - At the end of page 3, the authors claim that "We hypothesize that this is due to the high degree of overlap between the information conveyed by node features X and the graph structure A", and then proceed with an "analysis" using the cartoons in Figure 1. First of all, I do not think that the hypothesis is correct in general. As a very simple example, consider a balanced stochastic block model with 2 equal-sized classes, inter-class edge probability q=0 and intra-class edge probability p=1. That is, consider a graph that consists of two complete subgraphs, with one subgraph corresponds to class 0 and the other subgraph corresponds to class 1. Furthermore, assign node IDs 1,2,...,N to the nodes uniformly at random, and let the node features be the one-hot encodings of node IDs. This makes I(X;A) = 0 since X and A are independent. However, it should be clear to see that, even in this case, a strong Laplacian regularization would make an MLP to perform nearly as good as, e.g., GCN. In my example above I am omitting details regarding the representation (which concerns with the ability to approximate/represent certain functions)  and generalization (which concerns with sample complexity) capabilities of MLP and GCN. I welcome the authors' feedback on this example. In any case, I would recommend the authors be careful about stating a hypothesis, in particular, when certain assumptions are required for a claim to be true.
  - I found that the authors were very vague in their discussion about comparing MLPs with GNNs. For example, in the second paragraph on page 4, when discussing Figure 1(b), the authors claim that "Thus, MLPs are expected to have the same capacity as GNNs in this context". The term capacity is a very vague term, here, do the authors refer to representational power, or do the authors refer to generalization capability? Note that when an MLP has a worse test accuracy than a GNN, it could be due to either expressiveness or generalization or both. Overall, I really could not appreciate the discussion in Section 3.1. I am not convinced by the authors' conclusion that the performance gap between MLP and GNN is determined by the mutual information. I would recommend the authors constrain the problem context by explicitly stating the required assumptions and focus on specific data generative models. Otherwise, the current claims feel very unsupported.
  - As another example, at the top of page 4, the authors claim that "..., a well-trained MLP model might implicitly derive an estimated graph structure matrix from the outset. It could then implicitly leverage both the estimated graph structure and the raw node features to generate node embeddings, thereby achieving performance similar to that of GNNs." This claim is neither theoretically justified nor empirically verified in this paper. Even in the extreme case where one considers augmented feature matrix X' = [X; A], so that the graph structure is completely covered by X', it is unclear if an MLP acting on X' can achieve similar performance to that of GNNs.
  - To be honest, I think the paper might be better if the authors simply delete Section 3.1. Alternatively, the authors should be very careful in the writing. Clearly state all necessary assumptions on the data/model/architecture, clearly define and use proper technical terms, and avoid using vague terms. Otherwise, I find it very hard to be convinced by the messages which the authors try to convey in Section 3.1.

- There is a clear gap between $H(A|X)$ and $H(A|\hat{A})$ in equation 1, and the paper does not provide a bound on $|H(A|X)-H(A|\hat{A})|$. Therefore, it is difficult to determine how useful the proposed metric $H(A|\hat{A})$ is. The authors should at least discuss the reason to choose the $\ell_2$ distance for $\hat{A} = f(X)$.

- Although the empirical results over small datasets in Section 4 look promising, the additional results over larger datasets in the appendix are not very good. Since one of the major advantages of MLPs over GNNs is efficiency, the performance of InfoMLP over larger datasets is more relevant and important. I think the authors should provide results on larger datasets in the main text, and place the results over small datasets in the appendix. However, over larger datasets, InfoMLP is outperformed by simple GNN architectures such as GCN or even SGC which does not suffer from efficiency issues.

**Questions:**

- In Theorem 1, why do we necessarily have that $H(A|X) = \inf_f H(A|\hat{A})$ (i.e. equality holds)? Based on the proof I think it should be $H(A|X) \le \inf_f H(A|\hat{A})$?

- For training and testing time of SGC reported in Table 5: I think that once preprocessing is done, SGC is basically a liner logistic regression. Why does it have slower training and testing time than InfoMLP?

- At the end of page 6, the authors claim that "Furthermore, the node embeddings $Z_{aug}$ are utilized to predict the node labels." Based on the description of the method, shouldn't $Z_{mlp}$ be utilized to predict the node labels? In any case, I think the authors should report the performance achieved by both $Z_{aug}$ and $Z_{mlp}$ for comparison purposes.

---

> ### Author Response · Authors · 2023-11-22
> **Response to Reviewer zabj  (1/3)**
>
> We thank the reviewer for the insightful comments and constructive suggestions. Below are our answers to the weaknesses and questions.
>
> > W1: Claims in Section 3.1 are not well justified.
>
> **The data generation process behind our assumption.**
>
> The conclusion in Section 3.1 is indeed based on an assumption of graph-structured data generation. Specifically, our hypothesis posits that for a graph $G = (X, A)$, its generation process involves generating raw node features X for the nodes, and the graph structure A is generated based on the node features and additional confounder factors. This assumption is adopted by the research in the study of generative models for graphs [1]. In addition, many real-world graphs can be considered to be generated based on this assumption. For example,
> - **Citation networks**: Consider citation networks where the node features are the raw text of papers, and the edges represent citation relationships. In this case, we can hypothesize that the textual content of papers is generated first. Then, based on the content of the papers and authors' actions, citation relationships are formed.
> - **Social networks**: Consider social networks where the node features are user profiles, including attributes like occupation and interests, and the edges represent friendship relationships. We posit that users are initially generated with their individual attributes. Subsequently, friendship relationships are formed among users based on shared interests, real-world interactions, and other factors.
>
> Therefore, our assumption regarding the generation process of graph data is reasonable. Based on this assumption, we provide the analysis in Section 3.1. We will provide a detailed explanation of this assumption in the revised paper.
>
> **The balanced stochastic block model.**
>
> The first example provided by the reviewer, on the one hand, does not align with the generation process we consider for graph-structured data. On the other hand, this synthetic dataset is too specific and may not represent real-world graph datasets. Furthermore, **in the examples provided by the reviewer, it cannot be demonstrated that MLPs can achieve the same performance as GNNs**. The reviewer seems to consider only the transductive setting, where all the nodes (including the testing nodes) are involved in the training process. However, in the inductive setting, once testing nodes are not involved in the training process, for a new testing node with features represented as one-hot IDs, the MLP model clearly cannot make correct predictions, while the GNN model can easily classify it based on which block it belongs to
>
> **We are comparing the generalization ability of MLP and GNN.**
>
> Regarding the second point, we are indeed considering the generalization ability of MLP, and we appreciate the reviewer's input. In fact, the expressive power of MLP can hardly be enhanced because its input information and MLP's structure are fixed. The reason why GNN performs so well in semi-supervised learning is largely due to its utilization of graph structural information to enhance its generalization ability to non-training data, as mentioned in recent literature [2].
>
> **The performance of $X' = [X, A]$ under MLP classifiers.**
>
> Thanks for the suggestions. We agree with the reviewer that our claim needs to be empirically verified. Therefore, we follow the reviewer's suggestions and try learning an MLP classifier over the augmented node features $X' = [X, A]$. In addition, we can consider other simpler examples. On the one hand, SGC can be seen as an MLP model that takes the convoluted node features as input, and in this case, SGC has achieved performance similar to GNN. On the other hand, we can also use node structural embeddings (e.g., derived from DeepWalk) as an additional input for the MLP. In the table below, we provide the performance using the reviewer's example $X' = [X, A]$, as well as the performance with structural embeddings as input for the MLP model:
>
> | Model                 | Cora | Citeseer | Pubmed | Arxiv |
> |----------|------------|--------------|---------|---------
> |   $X' = [X, A]$    |       $81.5\pm0.4$   |  $71.9\pm0.3$  | $79.9\pm0.4$  | OOM   |
> |  Structural embedding    |   $82.2\pm0.5$    | $71.7\pm0.4$  | $80.5\pm0.5$  | $70.28\pm0.19$   |
>
> As presented in the table, various approaches that use structural information as input for MLPs can enable them to achieve performance comparable to GNNs.
>
> We appreciate the constructive criticism provided by the reviewer regarding Section 3.1 of our paper. The feedback has allowed us to better introduce the background and implications of our research. We hope that our response addresses your concerns.
>
> References:
>
> [1] Ma, Jiaqi, et al. "A flexible generative framework for graph-based semi-supervised learning." NIPS 2019.
>
> [2] Yang, Chenxiao, et al. "Graph Neural Networks are Inherently Good Generalizers: Insights by Bridging GNNs and MLPs." ICLR. 2023.

---

> ### Author Response · Authors · 2023-11-22
> **Response to Reviewer zabj  (2/3)**
>
> > W2: Gap between $H(A|X)$ and $H(A|\hat{A})$
>
> The gap between $H(A|X)$ and $H(A|\hat{A})$ is determined by how the function $\hat{A} = g(X)$ is formulated. Since $|H(A|X) - H(A|\hat{A})| = I(X;A|\hat{A}) = I(X;A) - I(X;A;\hat{A}) = {\rm const} - I(A;\hat{A})$. We cannot provide a specific bound because the constant value $I(X;A)$ is non-tractable. Luckily, since we do not care about the absolute value between the gaps, we can resort to the mutual information between the original graph adjacency matrix and the estimated graph structure matrix $\hat{A}$, i.e., $I(A;\hat{A})$ for reference. To be detailed, to make this bound tighter, we only need the estimated adjacency matrix to be very close to the true adjacency matrix.
>
> We choose the $\ell_2$ distance primarily motivated by the well-known graph homophily theory, which suggests that nodes with similar features are more likely to be connected than nodes with dissimilar features. Therefore, the $\ell_2$ distance between two nodes based on their features can, to some extent, be used to infer whether two nodes are connected.
>
> The greatest advantage of this approach is that $g(X)$ is non-parametric, so it can be used as a plug-and-play method for any dataset without the need to train a separate model. Even with such a simple formulation, it is already sufficient to capture the graph structure information and the correlation between node features in graph datasets. It can also explain why MLP-typed models exhibit drastically different performance on different graph datasets, as presented in Table 2 and Figure 2. For instance, on datasets with smaller $H(A|\hat{A})$ (e.g., Cora, Citeseer, Pubmed, and Coauthor-CS), the performance of MLP-typed models is evidently closer to powerful GNN models than datasets of larger $H(A|\hat{A})$ (e.g., Amazon-Computer).
>
> > W3: Empirical results on large graphs are not satisfying
>
> Thank you for carefully checking our results on the large graphs. Our response is divided into two parts: 1) The suboptimal performance of InfoMLP on some large-scale graphs is due to the significant information gap between node features $X$ and the graph adjacency matrix $A$. 2) MLP-typed methods have an unparalleled advantage over GNNs in cold-start scenarios.
>
> **The suboptimal performance of InfoMLP on some large-scale graphs is due to the significant information gap between $X$ and $A$.**
>
> Critically, the suboptimal performance of InfoMLP is not caused by the large graph size but by the information gap between node features and graph structures, which is endemic to the quality of data. At the end of Section 3.2, we have concluded three representative cases of the scale of $H(A|\hat{A})$. The third case (large $H(A|\hat{A})$) has pointed out that when the node features $X$ fail to contain adequate information about the graph structure $A$, the performance of the MLP model can hardly be improved to the level of GNNs. This is the common case for large-scale graphs since $N\times N \gg N \times d$ when $N$ is large while $d$ is small. For example, Arxiv is a citation network with over 160k nodes, while the node attributes consist of 128-dimensional encoded features from raw textual features using skip-gram. The resulting information gap endemic to the data hampers the ability to train an MLP model with enough expressive power comparable to GNNs.
>
> The empirical results in Figure 5, Appendix E.4, in the original paper also support our arguments. As depicted in Figure 5, the pos/neg distributions are heavily overlapped, leading to large conditional entropy $H(A|\hat{A})$. It is precisely because the node features $X$ do not contain enough information to recover the graph structure $A$ that MLP-typed methods fail to compete with GNNs
>
> **MLP-typed methods have an unparalleled advantage over GNNs in cold-start scenarios.**
>
> As explained in the second paragraph of Sec. 1, our study is motivated by addressing the two-fold limitations of conventional GNNs: 1) the efficiency issues and 2) the challenges in achieving good performance in cold-start scenarios. While it is true that shallow GNNs can partially alleviate the first issue, they are less preferable than MLPs in cold-start scenarios. We conducted additional experiments on Reddit and Arxiv datasets for cold-start scenarios and presented the results in the table below.
>
> |  Cold-start | Reddit | Arxiv |
> | ------- | ------- | ------- |
> | GCN | $59.7\pm1.2$  | $56.7\pm0.5$ |
> | InfoMLP | $66.3\pm1.4$ | $58.2\pm0.4$ |
>
> Compared to the results in the transductive scenario, GCN experiences a significant performance drop in the cold-start scenario, falling far behind our InfoMLP. So, we believe that even though our proposed InfoMLP may not be as competitive as GNNs in some datasets in the transductive scenario, its advantages in the cold-start scenario demonstrate its value.

---

> ### Author Response · Authors · 2023-11-22
> **Response to Reviewer zabj (3/3)**
>
> > Q1: Theorem 1: Why the equality can hold.
>
> If $\hat{A} = f(X)$ is a fully free function, the equality can be obtained when $H(X|\hat{A}) = 0$, since we can simply let $\hat{A} = X$ (when $N = d$) or let $\hat{A} = {\rm concat}(X, 0)$ (when $N > d$). We understand the reviewer's concern might be that the above formulation of $\hat{A}$ is very special and unusual, while we think $H(A|X) = \inf\limits_{f}H(A|\hat{A})$ is more accurate. (The equation suggested by the reviewer is correct as well)
>
> > Q2: Testing time of SGC
>
> The training/testing time of SGC includes the graph convolution step and, therefore, is larger than other MLPs. If we view the graph convolution as a preprocessing step (which is reasonable), the training/testing time will be the same as MLPs. We've revised it in the updated paper, and thanks a lot for pointing it out.
>
>
> > Q3: Comparison of the performance of $Z_{\rm aug}$ and $Z_{\rm mlp}$
>
> Thank you for the valuable comments. The sentence you pointed out is a typo, and has been revised to be "Furthermore, the node embeddings $Z_{\rm mlp}$ are utilized to predict the node labels".
> In fact, $Z_{\rm aug}$ cannot be directly compared to $Z_{\rm mlp}$ because in our model, the cross-entropy loss for downstream classification tasks is applied to $Z_{\rm mlp}$ rather than $Z_{\rm aug}$. To meet the reviewer's request, we independently trained a classification model based on $Z_{\rm aug}$. In the table below, we provide the performance obtained with $Z_{\rm aug}$ in the transductive setting and compare it to $Z_{\rm mlp}$.
>
> |  Embedding | Cora  | Citeseer | Pubmed |
> | ------- | ------- | ------- |  ------- |
> | $Z_{\rm aug}$ | $81.9\pm0.5$  | $72.5\pm0.5$ |  $80.2\pm0.4$ |
> | $Z_{\rm mlp}$ | $83.8\pm0.3$ | $73.7\pm0.3$ | $83.2\pm0.9$ |
>
> As we can see, $Z_{\rm aug}$ generally performs worse on these datasets compared to $Z_{\rm mlp}$. We suspect that this is because the regularization loss employed by InfoMLP not only helps $Z_{\rm mlp}$ learn graph structural information but also includes a decorrelation loss, which encourages different dimensions of the embedding to be relatively independent. This, in turn, helps $Z_{\rm mlp}$ to obtain an embedding space that is more distinguishable and beneficial for classification.
>
> Furthermore, since $Z_{\rm aug}$ itself utilizes graph structural information as an input to the encoder, it is natural that it can achieve good performance when node features alone do not contain sufficient graph structural information. For example, on the Arxiv dataset, $Z_{\rm aug}$ effortlessly achieved a classification accuracy of $70.82\pm 0.16$.

---

> > ### Comment · Reviewer_zabj · 2023-11-22
> >
> > I thank the authors for providing a detailed response and carrying out additional experiments. However my biggest concern with respect to the analyses in this paper remains. Here are some suggestions:
> > - The authors should clearly state all the required assumptions with respect to the data generation process and the learning setting. For example, indicate that the paper assumes homophily in both graph structure and node features, that is, nodes from the same class not only have similar features as measured by Euclidean distance, but also are more likely connected with one another. In addition, indicate if the paper focuses on the transductive setting or the inductive setting or both. These contexts are not very clear in the current paper. It's difficult to judge the soundness of the claims without a clear context. For example, MLP can be better than a simple GCN if the node features are strongly homophilous but the graph structure is complete noise.
> > - Based on the author's response, the equality $H(A|X) = \inf_f H(A|\hat{A})$ does not hold in general. Please use $\le$ instead. The authors should discuss the gap between  $H(A|X)$ and $\inf_f H(A|\hat{A})$, or better, provide an upper bound on $H(A|\hat{A})-H(A|X)$ when $f$ is the $\ell_2$ metric.

---

> > > ### Author Response · Authors · 2023-11-23
> > > **Paper has been revised according to the reviewer's suggestions**
> > >
> > > We appreciate the reviewer's prompt response. Based on your suggestions, we have made significant revisions to Section 3 of our paper to provide a clearer explanation of the assumptions underlying this study, the research settings, and to offer a more rigorous analysis. Our revisions are summarized as follows:
> > >
> > > 1. We have added a separate preliminary section at the beginning of Section 3, introducing the definition of the problem studied in this paper and the settings considered (both transductive and inductive).
> > > 2. In Section 3.1, we provide a detailed explanation of the assumption underlying the analysis in this paper (see Assumption 1). This assumption is based on the generation process of graph data, including node features X, graph structure A, and node labels Y. We establish a connection between this assumption and the different sources of information that MLP and GNN rely on for predictive tasks, resulting in different outcomes.
> > > 3. In Section 3.2, we introduce the motivation and assumptions behind using the $\ell_2$ distance to infer graph structure. We also derive a bound for the gap between $H(A|X)$ and $H(A|\hat{A})$ when using the $\ell_2$ distance to infer graph structure (see Theorem 2). Theorem 2 further  demonstrates that when the difference between the distributions of positive edges and negative edges obtained using the $\ell_2$ is greater, the gap between $H(A|X)$ and $H(A|\hat{A})$ becomes smaller
> > >
> > > Thank you once again for your constructive suggestions. Your feedback has significantly improved the completeness and readability of our paper. Considering that the latest revised paper has incorporated your suggestions, we would greatly appreciate it if you could consider raising your score.

---

### Official Review · Reviewer_YPT6 · 2023-10-31

**Soundness:** 2 fair
**Presentation:** 3 good
**Contribution:** 2 fair
**Rating:** 3
**Confidence:** 4

**Summary:**

This paper addresses the question of how to incorporate structural information to MLPs.
In other words, the latent question is: How structured must be MLPs?
Quoting the authors: "In this work, we first aim to understand the reasons behind the successes
of previous MLP-structured models for learning node representations".
The main idea is to quantify the overlap between the node features and graph structures (it is
well known that MLPs perform better than GNNs when the node feaures are uncorrelated with
the graph structure/adjacency). As a result, the authors propose to maximize the mutual information
between node embeddings and adjacency as a pre-processing step.

InfoMLP, the proposed method, works as follows: "1) the generation of a graph-augmented node
feature matrix that encapsulates extensive graph structure information, and 2) the maximization of
mutual information between node representations learned from the original node features and the
generated graph-augmented node features.".

Conditional probability between existing edges and feature correlations feeds an entropy estimator
that proves to illustrate the overlap between feature distribution and structure (links) in well-known
datasets. This leads to an upper bound of the H(A|X): conditional entropy of A given X.

Then, 1) is addressed in order to facilitate information maximization. This leads to $X_{aug}$ as a
function of powers of the transition matrix. 2) is addressed through standard MI maximization.
However, for the sake of efficiency, the MI Loss during training is simply an Euclidean loss. Overall,
"InfoMLP has the same complexity as a vanilla MLP in both training and testing".

Experiments show that InfoMLP is good in cold-start settings and outperforms significantly
the state-of-the-art only in Pubmed. It fails in Cora.

**Strengths:**

* Nice methodology for comparing/integrating MLPs-GNNs.
* Good analysis of mutual information.

**Weaknesses:**

* Not good results in standard datasets (e.g. Cora).
* This is probably due to the terse definition of the augmented features. Multi-hop diffusion seems to me not sufficient even when $K$ is adapted properly.

**Questions:**

* How critical is the augmented representation?
* How critical is the MI simplified Loss? (please see Rényi estimators).
* Entropy estimation is very terse (positive vs negative distribution). Please, measure the overlap between positive and negative distribution using the Chernoff bound, for instance.

---

> ### Author Response · Authors · 2023-11-22
> **Response to Reviewer YPT6 (1/3)**
>
> We appreciate the reviewer's insightful comments and suggestions for improvement. However, we also noticed that some of the questions might arise from a potential misunderstanding of the setting studied by our paper. We address these points in the following answers.
>
> > W1 & W2: Not good results in standard datasets (e.g. Cora).
>
> We'd like to first clarify upfront the three different experimental settings we considered, where in each case, the models should be compared in different manners given their different input information.
>
> > 1) In the ***transductive setting***, all nodes and edges are observed during both the training and testing phases.
> > 2) In the ***inductive setting***, testing nodes and their related edges are unobserved during training but are accessible when testing.
> > 3) ***Cold-start setting*** is an extreme case of the inductive setting, wherein the edges related to the testing nodes are inaccessible during testing as well. (We call it cold-start setting since this is similar to the cold-start recommendation problem where the historical interactions are inaccessible for new users)
>
> These different settings pose different levels of learning difficulties for GNN-based and MLP-based approaches since both model classes use different information as input. Let $X_{train}$ and $X_{test}$ be the training/testing node attributes, and let $E_{train}$ and $E_{test}$ be the edges related to the training and testing do, respectively. The following tables compare the observed information for two model classes.
>
> |  |Transductive GNN  | Transductive MLP  |
> | ------- | ------- | ------- |
> | Train | $X_{\rm train}, X_{\rm test}, E_{\rm train}, E_{\rm test}$ | $X_{\rm train}, X_{\rm test}, E_{\rm train}, E_{\rm test}$ |
> | Test | $X_{\rm train}, X_{\rm test}, E_{\rm train}, E_{\rm test}$ | $X_{\rm train}, X_{\rm test}, E_{\rm train}, E_{\rm test}$ |
>
> |  |Inductive GNN  | Inductive MLP  |
> | ------- | ------- | ------- |
> | Train | $X_{\rm train}, E_{\rm train}$ | $X_{\rm train},  E_{\rm train}$ |
> | Test | $X_{\rm train}, X_{\rm test}, E_{\rm train}, E_{\rm test}$ | $X_{\rm train}, X_{\rm test}, E_{\rm train}$ |
>
> |  |Cold-start GNN  | Cold-start MLP  |
> | ------- | ------- | ------- |
> | Train | $X_{\rm train}, E_{\rm train}$ | $X_{\rm train}, E_{\rm train}$ |
> | Test | $X_{\rm train}, X_{\rm test}, E_{\rm train}$ | $X_{\rm train}, X_{test}, E_{\rm train}$ |
>
> Based on the experimental settings, we next illustrate how our achieved results serve to validate the effectiveness of our model.
>
> > Table 3 in our paper presents the results for **transductive setting** (where MLP and GNN use the same information), and the proposed InfoMLP significantly outperforms all GNN baselines in such a setting.
>
> > Table 4 in our paper reports the results for **inductive and cold-start settings**. Please notice the differences between these two settings. In the inductive setting, GNNs can leverage additional information ($E_{\rm test}$ during the testing phase) over MLP-based models, which allows GNN to utilize the connection information of testing nodes. Hence, it is natural for GNN to outperform the MLP model in the inductive setting. But for cold-start settings, as shown in Table 4, our proposed InfoMLP yields significant improvements.
>
> To sum up, when both model classes use the same input information and are compared in a fair setting, our InfoMLP exhibits clear superiority.

---

> ### Author Response · Authors · 2023-11-22
> **Response to Reviewer YPT6 (2/3)**
>
> > Q1: How critical is the augmented representation?
>
> The graph-augmented node feature matrix $X_{\rm aug}$ plays an important role in the proposed InfoMLP, since we aim to minimize $H(A|X_{\rm aug})$ such that $X_{\rm aug}$ contains enough information about the graph structure. In our paper, we assume $X_{\rm aug} = g(X, A) = \sum\limits_{k=1}^K \gamma_k \tilde{A}^{k}X$, and for simplicity we let $\gamma_k = 1/K$. In this way, we only need to tune $K$ to obtain a proper $X_{\rm aug}$.
>
> In Figure 3 of the original paper, we studied the impacts of different instantiations of $X_{\rm aug}$ on InfoMLP's performance by adjusting $K$. Our observation is that there is a positive correlation between InfoMLP's performance and $H(A|X_{\rm aug})$ estimated by our estimators. Additionally, we observe that when $K=1$, $H(A|X_{\rm aug})$ is in general, larger (and InfoMLP has suboptimal performance). However, as $K$ increases to 2 and beyond, $H(A|X_{\rm aug})$ quickly converges to smaller values, and at this point, InfoMLP's performance also approaches the optimum. This indicates that, for the graphs studied in Figure 3, combining the information of 2-5 order neighborhoods with node attributes is sufficient for $X_{\rm aug}$ to reconstruct the original graph structure $A$.
>
> In addition, we supplement more discussions on the impact of $\gamma_k$ on the performance of InfoMLP. We let $\gamma_k = \alpha(1-\alpha)^k$ such that the definition of $X_{\rm aug}$ recovers the classical Personalized PageRank embeddings[1]. This specification leads to the property that with increasing $K$, the weight of each hop is $(1-\alpha)$ times the weight of the previous layer, thereby gradually reducing the importance of higher-order neighbors when $0 < \alpha < 1$. We let $\alpha_k = 0.2$, and present the performance of InfoMLP on Cora dataset in the transductive setting w.r.t. to $K$ in the table below.
>
> |  K | 1 | 2 | 3 | 4 | 5 | 6 | 7 |
> | ------- | ------- | ------- |------- | ------- | ------- |------- | ------- |
> | InfoMLP | $80.2\pm 0.6$ | $82.5\pm 0.6$ | $83.1\pm 0.5$ | $83.5\pm 0.5$ | $83.6\pm 0.5$ | $83.5\pm 0.5$ | $83.5\pm 0.5$
>
> We have similar observations and results as in Figure 3. This indicates that InfoMLP can adapt to different instantiations of $X_{\rm aug}$.
>
> > Q2: How critical is the MI simplified loss?.
>
> Thanks for the nice suggestion that can help to improve our work. The Rényi estimator is indeed an important method for measuring mutual information between two variables. However, as shown in Equation 3, our learning objective is to maximize $I(Z_1, Z_2)$, which will be used to update the model's trainable parameters. This requires the MI estimator to be differentiable for backpropagation. Unfortunately, the Rényi estimator does not meet this requirement, making it unsuitable for our context.
>
> In addition to the loss used in this paper, based on the reviewer's suggestion, we supplement new experiments with the InfoNCE MI estimator [2] and the Mutual Information Neural Estimator (MINE) [3]. They have both been proven to have lower bounds on MI[4], making it suitable as the learning objective (the MI can maximized by maximizing the estimator). In the following table, we present the mean accuracy of InfoMLP with these estimators:
>
> |  MI Estimator | Cora | Citeseer | Pubmed |
> | ------- | ------- | ------- |------- |
> | MINE | $82.7\pm 0.9$ | $71.9\pm 0.8$ | $80.8\pm0.9$ |
> | InfoNCE | $84.0\pm 0.4$| $73.4\pm0.4$ | $82.9\pm0.3$ |
>
> The InfoNCE MI estimator provides competitive performance as the feature-decorrelation-based loss used in our paper (even better on some datasets), while MINE in general, yields inferior performance. This is because MINE has a very large variance in MI estimation, which results in less accurate estimates (see discussions in [4]). Although InfoNCE provides strong results, its computational complexity is O(N^2d), which makes it time and memory-consuming to calculate the InfoNCE loss on medium-sized graphs. Considering these, we use the estimator, which is efficient and produces competitive results in our paper.
>
>
> References:
>
> [1] Gasteiger, Johannes, Stefan Weißenberger, and Stephan Günnemann. "Diffusion improves graph learning." Advances in neural information processing systems 32 (2019).
>
> [2] Oord, Aaron van den, Yazhe Li, and Oriol Vinyals. "Representation learning with contrastive predictive coding." arXiv preprint arXiv:1807.03748 (2018).
>
> [3] Belghazi, Mohamed Ishmael, et al. "Mutual information neural estimation." International conference on machine learning. PMLR, 2018.
>
> [4] Poole, Ben, et al. "On variational bounds of mutual information." International Conference on Machine Learning. PMLR, 2019.

---

> ### Author Response · Authors · 2023-11-22
> **Response to Reviewer YPT6 (3/3)**
>
> > Q3: Entropy estimation is very terse (positive vs negative distribution).
>
> Thank you for the constructive suggestion. To ensure our results are interpreted in a precise way, we'd like to point out the difference between the estimated values of $H(A|\hat{A})$ (in the titles of Figure 2) and the divergence of positive/negative edge distribution. **The calculation method of the conditional entropy $H(A|\hat{A})$ used in this paper is rigorously derived through the definition of conditional entropy (see Equation 1 and Appendix D in the submitted paper)**. It is merely in the derivation process that we found its calculation is closely related to the positive/negative edge distributions. Therefore, the plotting of distributions in Figure 2 is only an illustrative representation.
>
> We agree with the reviewer's suggestion that computing the numerical metrics for measuring the distance between two distributions can add value to our paper. Therefore, we add new experiments calculating the relative entropy (KL-divergence) to measure the difference between the distributions of positive edges and negative edges. The table below shows the difference between the distributions of positive edges and negative edges measured by KL-divergence.
>
> |  Divergence | Cora | Citeseer | Pubmed |  Computer  | CS |
> | ------- | ------- | ------- |------- |------- | ------- |
> | KL-Divergence  | 0.4843 | 1.0093| 0.7928 |  0.1462 | 1.3617 |
>
> A larger KL-divergence value indicates a more significant difference between the two distributions. As expected, the KL-divergence values exhibit a clear negative correlation with the estimated values of $H(A|\hat{A})$ in Figure 2, which is consistent to our results. E.g., Computer dataset has the largest $H(A|\hat{A})$ while the smallest $\mathcal{D}\_{kl}(A|\hat{A})$. By contrast, CS dataset has the smallest $H(A|\hat{A})$ while the largest $\mathcal{D}_{kl}(A|\hat{A})$.

---

### Official Review · Reviewer_o93U · 2023-11-01

**Soundness:** 4 excellent
**Presentation:** 3 good
**Contribution:** 4 excellent
**Rating:** 8
**Confidence:** 3

**Summary:**

This paper studies an Information-Theoretic Perspective on node classification on graphs, specifically, it focuses on the problem of when and why MLP can achieve good performances in graph representation learning tasks. The conclusion drawn from this paper is that when there is significant structural information contained within node features, MLP can achieve relatively good performances, since it does not require additional operators to introduce topological information. Motivated by this finding, the authors proposed a solution named InfoMLP, which aims to learn a node embedding that maximizes the mutual information between node feature and graph structure.

**Strengths:**

1. The most notable contribution of this paper is it explained how and why MLP can achieve relatively good performances in node classification on certain graph datasets, the mutual information justification is intuitive and serves as a strong argument.

2. Based on the observation that, MLP can only achieve good performances when graph structure information is partially contained in node features, authors proposed a mutual-information maximization approach to enhance the performance of MLP on the graph, which is novel and interesting.

3. The proposed method exhibits strong improvements over baselines, notably, this improvement is valid on both homophilous and heterophilic graphs.

**Weaknesses:**

1. Despite the insights and meaningful findings of this paper, the information-theoretic perspective is still rather empirical (solely based on observations), and lacks theoretical guarantee.

2. What are the differences between $Z_{aug}$ and $Z_{mlp}$? The term "ideal" node representation is vague, a more rigorous definition might be needed here.

3. The maximization of $H(A|X)$ is not clear, I can only see a non-parametric mapping that somehow fuses the representation of $X$ and $A$ together into a single representation. However, there is no guarantee nor intuitive explanation of why and how $X_{aug}$ maximizes $H(A|X)$. If no theoretical justification can be provided, authors should try to explain this from an intuitive standpoint, supported by an ablation study that can illustrate or verify the mutual information contained within $X_{aug}$.

**Questions:**

No questions at the moment.

---

> ### Author Response · Authors · 2023-11-22
> **Response to Reviewer o93U**
>
> We thank the reviewer for the constructive comments and insightful suggestions.
>
> > What are the differences between $Z_{aug}$ and $Z_{mlp}$? The term "ideal" node representation is vague; a more rigorous definition might be needed here.
>
> **Differences between $Z_{\rm aug}$ and $Z_{\rm mlp}$**
>
> $Z_{\rm mlp}$ and $Z_{\rm aug}$ are the embedding matrix outputted from the same encoder MLP model, while with the different input information. Specifically, let ${\rm MLP}\_{\theta}$ be the MLP encoder parameterized by $\theta$, $Z_{\rm mlp}$ is ${\rm MLP}\_{\theta}$'s output using the raw node features as input, e.g., $Z_{\rm mlp} = \rm{MLP}\_{\theta} (X)$, while $Z_{\rm aug}$ is ${\rm MLP}\_{\theta}$'s output using the graph-augmented node features as input, e.g., $Z_{\rm aug} = \rm{MLP}\_{\theta} (X_{\rm aug})$. Since $X_{aug}$ is defined as a function of both the vanilla node features and the graph adjacency matrix, i.e., $X_{\rm aug} = g(X, A)$, $Z_{\rm aug}$ naturally encodes the graph structure information into the embeddings, while $Z_{\rm mlp}$ does not.
>
>
> **A more rigorous definition of "ideal" node representations**
>
>
> The term "ideal" node representations in the beginning of Section 3.3 refers to the node representations that, given the conditions presented in Figure 1, would perform satisfactorily in downstream tasks. To improve the clarity, we have updated the description "the node representations that are expected to perform well in downstream tasks given the condition depicted in Figure 1" in the revised paper.
>
>
> >The maximization of $H(A|X)$ is not clear, I can only see a non-parametric mapping that somehow fuses the representation of $X$ and $A$ together into a single representation. However, there is no guarantee nor intuitive explanation of why and how  $X_{\rm aug}$ maximizes $H(A|X)$.
>
> Thanks for the question. The reviewer should be referring to the minimization of $H(A|X_{\rm aug})$ (rather than maximization). The minimization of $H(A|X_{\rm aug})$ is achieved via tuning the optimal $\gamma_k$ and K in Equation 2. We provide a detailed explanation below.
>
> Since $X_{aug}$ is defined as the function output of the vanilla node features $X$ and the graph adjacency matrix $A$, i.e., $X_{\rm aug}(X, A)$, we can formulate the optimization problem as
> $$
>     \min\limits_{X_{\rm aug}} H(A|X_{\rm aug}) = \min\limits_{g}H(A|g(X,A)).
> $$
> However, the search space for $g$ is very large, as $g$ can be any function. Therefore, we choose to reduce the searching space for g using the form of Eq. 6, which is the popular graph diffusion mechanism for aggregating multi-hop neighborhood information.
> $$
>     g(X, A) = X_{\rm aug} = \sum\limits_{k=1}^K \gamma_k \tilde{A}^kX .
> $$
> In this way, the optimization problem is transformed into selecting the optimal $K$ and $\gamma_k$ such that $H(A|X)$ is minimized, which is much easier to solve.

---

> > ### Comment · Reviewer_o93U · 2023-11-22
> >
> > Dear authors, thanks for addressing my questions. Overall, my side of the questions has been concluded.
> >
> > However, after reading other reviewers' comment, I feel that reviewer zabj and YPT6 are perhaps more knowldegable in this area, I recommend prioritizing the addressing of their concerns. Currently, I will maintain my existing score and confidence level. However, upon the satisfactory resolution of the questions raised by these other reviewers, I am open to considering an increase in the confidence score.

---

### Author Response · Authors · 2023-11-23
**Response to all reviewers**

We would like to thank the reviewers for their time and effort spent evaluating our work and their constructive comments. We find the suggestions to be very helpful in improving the quality of our work, making it clearer and more convincing.

We have uploaded the revised version of the paper, in which we have made extensive modifications to Section 3 to make our claims clearer and the analysis more rigorous. We have also added more experimental studies in response to the reviewers' suggestions. We hope that these improvements will enhance the quality of this paper and address the concerns raised by the reviewers.

Thank you once again for your thorough review.

Best regards,

The Authors

---

### Meta-Review · Area_Chair_iUiQ · 2023-12-10

**Metareview:**

This paper first makes the observation that, while MLP is much more computationally efficient than GNN, its performance is comparable to that of GNN only if node representations include information of the graph structure. (This seems obvious since in general more information lead to more accurate predictions, so MLP works comparable to GNN only if its input can somehow include some graph structure information.) Based on this observation, InfoMLP is proposed in order to learn node representation with maximum mutual information with the graph structure.

**Justification For Why Not Higher Score:**

The proposed method lacks support from neither theoretical justification nor experimental results. The theory is vague with doubtful assumptions (such as using a normal distribution to model edge distances, which are always nonnegative). Experiments do not show improved results in some of the data sets.

**Justification For Why Not Lower Score:**

N/A

---

### Decision · Program_Chairs · 2024-01-16

Reject